# Peculiar COVID-19 effects in the Greater Tokyo Area revealed by spatiotemporal variabilities of tropospheric gases and light-absorbing aerosols

Alessandro Damiani[1], Hitoshi Irie[1], Dmitry A. Belikov[1], Shuei Kaizuka[1], Hossain Mohammed Syedul Hoque[2], Raul R. Cordero[3]

[1]Center for Environmental Remote Sensing (CEReS), Chiba University, Chiba, 2638522, Japan
[2]Graduate School of Environmental Studies, Nagoya University, Nagoya, 4640064, Japan
[3]Department of Physics, Universidad de Santiago de Chile, Santiago, 3363, Chile

*Correspondence to*: Alessandro Damiani (alecarlo.damiani@gmail.com)

**Abstract.** This study investigated the spatiotemporal variabilities in nitrogen dioxide ($NO_2$), formaldehyde (HCHO), ozone ($O_3$), and light-absorbing aerosols within the Greater Tokyo Area, Japan, the most populous metropolitan area in the world. The analysis is based on total tropospheric column, partial tropospheric column (within the boundary layer), and *in situ* observations retrieved from multiple platforms and additional information obtained from reanalysis and box model simulations. This study mainly covers the 2013–2020 period, focusing on 2020, when air quality was influenced by the coronavirus 2019 (COVID-19) pandemic. Although total and partial tropospheric $NO_2$ columns were reduced by an average of about 10% in 2020, reductions exceeding 40% occurred in some areas during the pandemic state of emergency. Light-absorbing aerosol levels within the boundary layer were also reduced for most of 2020, while smaller fluctuations in HCHO and $O_3$ were observed. The significantly enhanced degree of weekly cycling of $NO_2$, HCHO, and light-absorbing aerosol found in urban areas during 2020 suggests that, in contrast to other countries, mobility in Japan also dropped on weekends. We conclude that, despite the lack of strict mobility restrictions in Japan, widespread adherence to recommendations designed to limit the COVID-19 spread resulted in unique air quality improvements.

## 1 Introduction

Fossil fuel combustion is the dominant source of nitrogen oxides ($NO_x$) in the atmosphere, to which traffic is the main contributor, followed by thermal power plants. Other sources include emissions from fires, soils, and lightning. Consequently, $NO_x$ are among the main drivers of air quality degradation in urban areas, and epidemiological studies have shown that nitrogen dioxide ($NO_2$) exposure is often associated with adverse health effects such as lung cancer, asthma, and cardiopulmonary mortality (e.g., Hamra et al., 2015; Achakulwisut et al., 2019). Therefore, $NO_2$ is carefully monitored using both surface and satellite instruments, and is often used as an indicator of air pollution. Moreover, $NO_x$ are precursors to secondary aerosols and catalyze the formation of tropospheric ozone ($O_3$), with consequences for the climate. Due to its short lifetime on the order of a few hours, $NO_2$ is an excellent marker for anthropogenic emissions, and satellites often show

enhanced $NO_2$ around large cities and thermal power plants (e.g., Beirle et al., 2003). Therefore, in the past, satellite $NO_2$ observations have been exploited to evaluate the effectiveness of long-term abatement strategies, the effects of economic recessions, and the impacts of short-term emissions regulations on air quality (e.g., Russell et al., 2012; Lee et al., 2021; Vohra et al., 2022).

Along with volatile organic compounds (VOCs), $NO_x$ are major $O_3$ precursors. Nevertheless, in contrast to rural sites where $NO_x$-limited conditions usually prevail, in urban locations, under VOC-limited (i.e., $NO_x$-saturated) conditions, a reduction in $NO_x$ is often associated with an increase in $O_3$ due to reduction of the NO titration effect (Murphy et al., 2007). Indeed, in recent years, satellite observations showed that, although $NO_x$ emissions are still rising in various developing countries (e.g., India), they have significantly decreased in the majority of the developed countries of North America, Europe, and East Asia

(Russell et al., 2012; Geddes et al., 2016; Georgoulias et al., 2019), while tropospheric ozone has increased (Ziemke et al., 2019; Li et al., 2019; Lee et al., 2021).This general tendency has been confirmed in Japan (e.g., Akimoto, 2017) and, recently, in the Kanto region, where both $NO_2$ and formaldehyde (HCHO) (as a proxy for non-methane VOC, NMVOC) were reduced and $O_3$ recovered slightly during the period of 2013–2019 (Irie et al., 2021).

Due to their association with human activities, anthropogenic $NO_x$ emissions often display a weekly cycle. The term

'weekend effect' refers to the difference in pollutant concentrations between weekdays and weekends (e.g., Cleveland et al., 1974). In the same manner, emissions usually decrease during holiday periods, and an equivalent 'holiday effect' can be defined (e.g., Tan et al., 2009). Reduced $NO_x$ levels on rest days are often coupled with increased $O_3$ in and around cities (Cleveland et al., 1974; Murphy et al., 2007) but not in rural areas (Sicard et al., 2020a). The weekly cycles of both $NO_2$ and $O_3$ have been extensively investigated in previous research using both ground-level (Cleveland et al., 1974; Sadanaga et al.,

2012; Zou et al., 2019; Sicard et al., 2020a) and satellite (Beirle et al., 2003; Stavrakou et al., 2020) observations. A satellite-based analysis of the global temporal evolution of the $NO_2$ weekly cycle showed that it has become smaller in recent years (Stavrakou et al., 2020). This trend has been explained by the current reduction in $NO_x$ emissions, confounding of anthropogenic and background $NO_x$, and secondarily, by an increase in $NO_2$ lifetime. On the other hand, a recent study of the $O_3$ weekend effect revealed a significant downward trend (Sicard et al., 2020a). Overall, analysis of the response of $O_3$

formation to emission changes supports the development of strategies to reduce precursor emissions and improve air quality.

The lockdowns caused by the coronavirus disease 2019 (COVID-19) pandemic represent an opportunity to assess the impacts of human activities on the environment and human health (Gkatzelis et al., 2021; Shakil et al., 2020; Laughner et al., 2021). Recently, many studies have shown that $NO_2$ levels in 2020 were lower than in previous years due to reduced

anthropogenic emissions associated with reduced mobility during temporary lockdowns (e.g., Bauwens et al., 2020; Zander et al., 2020; Cooper et al., 2022). Moreover, these $NO_2$ reductions in 2020 were associated with surface $O_3$ enhancement (Sicard et al., 2020b) and reduced $O_3$ levels in the free troposphere (Steinbrecht et al., 2021; Miyazaki et al., 2021). Increases in surface $O_3$ in some cities were explained by a reduction of the NO titration effect (Sicard et al., 2020b).

After the first COVID-19 cases in Wuhan, to prevent the spread of the pandemic, strong social distancing and quarantine measures were implemented in many Chinese cities as early as 24 January 2020 till about 25 February; then, measures gradually downgraded to a partial lockdown. Evident decreases in most air pollutant concentrations have been reported for China, with satellite-based $NO_2$ reductions of about 40 % (Bauwens et al., 2020; Le et al., 2020). In February, TROPOMI $NO_2$ and HCHO decreased to about 83% and 11% in Wuhan, respectively (Ghahremanloo et al., 2021). Moreover, surface observations showed a general increase in surface ozone in most of the regions, although ozone decreased in the subtropical south and, besides the reduced emissions, meteorological changes were found to be important contributors (Sicard et al., 2020b; Le et al., 2020; Liu et al., 2021). In Korea, the most significant changes occurred in March, with a reduction of about 20% in $NO_2$ and 45% in PM2.5 nationwide, while surface ozone, in contrast with China, was slightly decreased (Ju et al., 2021).

Overall, data from various sources suggest a decline in worldwide mobility in 2020 (Nouvellet et al., 2021). Figure 1 shows the change in mobility (amount of people transiting public stations, derived from Google), compared to the pre-COVID period, for some large metropolitan areas around the world deeply affected by the pandemic (further details in Section 2.1.7). The larger reductions (with peaks over 90%) generally coincided with lockdowns, which occurred at slightly different times in different countries. Outside of lockdowns, mobility remained markedly decreased throughout 2020.

In Japan, the change in mobility was smaller and more gradual compared to other countries due to the avoidance of strict legal restrictions. However, on February 3, Japan began addressing the issue of the Diamond Princess cruise ship quarantine and on March 3, primary and secondary schools were closed and replaced with remote learning (Table 1). Then, an official state of emergency, when the most stringent restrictions were in effect, was declared from April 7 to May 25. Thereafter, the situation never returned to normal, as new periods of reduced mobility coincided with the second wave of coronavirus infections, which peaked in July–September, and the third wave starting in December.

Table 1. Milestones of the COVID-19 pandemic in Japan.

| Date | Milestone |
| --- | --- |
| Jan. 15, 2020 | First COVID-19 infection |
| Feb. 3, 2020 | Diamond Princess cruise ship quarantine |
| Feb. 13, 2020 | First COVID-19 death |
| Mar. 3. 2020 | Closure of schools |
| Mar. 11, 2020 | COVID-19 declared pandemic by WHO |
| Mar. 24, 2020 | Tokyo Olympic Games postponed |
| Apr. 7 to May 25, 2020 | State of emergency |
| Jul. 22, 2020 | Start of "Go To Travel" program |

| Jan. 7 to Mar. 7, 2021 | State of emergency |
|---|---|
| Feb. 17, 2021 | Start of COVID-19 vaccinations |
| Jul. 12 to Sep. 30, 2021 | State of emergency |
| Jul. 23 to Aug. 8, 2021 | Tokyo 2020 Olympic Games |
| Aug. 24 to Sep. 5, 2021 | Tokyo 2020 Paralympic Games |

Generally, the mean reduction in mobility was similar among all weekdays and slightly smaller during the weekend in most
of the examined countries (Fig. 1b). In contrast to this worldwide trend, Japan showed the largest mobility drop during the
weekend, with decreases about 10% larger than on weekdays. Although limited to the state of emergency period and the
Tokyo area, this behavior has been observed from other data sources (e.g., Figure 2 in Sugawara et al., 2021) and appears to
be a robust feature representing common Japanese habits modified by the spread of the pandemic.

Due to this anomalous change in mobility, we suggest that the (relative) $NO_2$ weekend effect in 2020, primarily driven by
traffic, may show peculiar characteristics in Japan. Potentially, this effect could be larger than usual, despite reduced $NO_x$
emissions tending to reduce the anthropogenic weekly signature (Stavrakou et al., 2020). This unique situation provides the
opportunity to examine the changes in the air quality that occurred in 2020 in Japan and compare them with previous years.

As detailed above, many previous studies examined COVID-related changes in air quality on a global to local scale.
Nevertheless, due to somewhat soft countermeasures to limit the spread of the pandemic adopted in Japan with consequent
more limited change in the mobility compared with other countries, relatively fewer studies focused on this area (e.g.,
Itahashi et al., 2022). In some cases, changes in relevant air-quality parameters observed by ground-based or satellite
instruments in the Tokyo center during the emergency period have been examined on a local scale (Sugawara et al., 2021) or
related to other cities/countries on a global scale (e.g., Cooper et al., 2022) within studies aimed at comparing such
variabilities with mobility changes. Nevertheless, as we will see, such changes hide a sizeable spatiotemporal variability and
a widespread adherence to recommendations designed to limit the spread of the pandemic, which caused modification of
common habits. Those resulted in a unique air quality signature not limited only to the emergency period, which should be
examined on a regional scale.

We focused our study on the Greater Tokyo Area (GTA), in the Kanto region, which is the largest area of flat land in Japan,
extending inland from the Pacific coast (Fig. 2). It is the most populous metropolitan area in the world and the most
important economic hub of East Asia, and local emissions dominate it. Most of this large urban area is expected to be under
VOC-limited conditions (Akimoto, 2017; Irie et al., 2021). Nevertheless, western Japan and, to a lesser extent, this region are
usually affected by transboundary air pollution from the continent (Itahashi et al., 2022). Due to the strict mobility restrictions
implemented in China, this additional contribution is expected to be reduced in early 2020 (Itahashi et al., 2022). This makes the

analysis of the COVID-related effects even more complex and points to the necessity of a regional study focusing on spatiotemporal variability.

In this study, we apply an integrated approach that exploits various independent datasets retrieved from multiple platforms, including observations of $NO_2$, $O_3$, and HCHO (as a proxy for non-methane VOC, NMVOC, Sillman, 1995) from two sites equipped with Multi-Axis Differential Optical Absorption Spectroscopy (MAX-DOAS) systems located in urban and suburban areas. Further, anthropogenic light-absorbing aerosol data are presented. In contrast to most previous studies, which have focused on in situ and satellite-based data, observations representative of the boundary layer (< 1 km) provide our reference data to link surface and satellite column observations.

## 2 Datasets and methods

### 2.1 Datasets

#### 2.1.1 MAX-DOAS

We used MAX-DOAS observations operated at Chiba University (in 2013-2020) and Tsukuba (in 2015-2020) sites, which are located in urban and semi-urban environments within the Kanto region (Fig. 2). The MAX-DOAS technique is based on the well-established DOAS technique (Platt and Stutz, 2008), which utilizes differential absorption structures of trace gases in ultraviolet (UV) and visible (VIS) wavelength regions to derive aerosol and trace gas information (Hönninger et al., 2004). Our MAX-DOAS system is equipped with a UV-VIS spectrometer located indoors, while an outdoor telescope unit collects scattered sunlight at reference and off-axis elevation angles. A set of scattered sunlight spectra was measured in a 15-min interval. High-resolution spectra were recorded from 310 to 515 nm using the Maya2000Pro spectrometer (Ocean Insight, Inc., Orlando, FL, USA) with a slit of 25 μm and a full width at half maximum of approximately 0.3–0.4 nm, embedded in a temperature-controlled box. Wavelength calibration was performed daily, using a high-resolution solar spectrum, to account for potential long-term degradation of the spectrometer. Retrieval was conducted based on DOAS and optimal estimation methods using the Japanese MAX-DOAS profile retrieval algorithm, version 2 (JM2) (Irie et al., 2011, 2015) for the following purposes: DOAS fitting, retrieval of the aerosol profile and retrieval of trace gases. In this study, partial vertical columns of $O_3$, $NO_2$, and HCHO concentrations below 1 km were used. Notably, due to the contribution of upper troposphere/lower stratosphere (UT/LS) $O_3$ to differential slant column densities, only data collected at a solar zenith angle (SZA) below 50° can be used for $O_3$ retrieval (Irie et al. 2011, 2021). Therefore, the ozone dataset employed here was limited to the period of March–October. For cloud screening, measurements with retrieved aerosol optical depth (AOD) greater than 3 and relative humidity over water greater than 90% were excluded (Takashima et al., 2009). The uncertainty in the retrieved profiles was further reduced by averaging the data from four collocated MAX-DOAS instruments pointing in four different directions at the Chiba site (see inset in Fig. 2). This procedure is expected to better account for the potential spatial heterogeneity of tropospheric gases. By contrast, observations from a single MAX-DOAS system were used for the Tsukuba site. The MAX-DOAS horizontal viewing distance depends on atmospheric conditions (i.e., aerosol load), and can

be up to 24 km in the lowest 1-km vertical layer. Therefore, the MAX-DOAS partial column observations at Chiba are representative of the boundary layer across a region larger than (or comparable to) a usual satellite pixel, but sampled at higher accuracy. Within this framework, it is worth to highlight that our MAX-DOAS observations contributed to recent efforts of the scientific community to validate TROPOMI $NO_2$ and HCHO datasets at a global scale (Verhoelst et al., 2021; De Smedt et al., 2021).

### 2.1.2 TROPOMI

The Tropospheric Monitoring Instrument (TROPOMI) onboard the European Union Copernicus Sentinel 5 Precursor (S5P) satellite has a Sun-synchronous orbit with a daily equator crossing time of approximately 13:30 local solar time and daily global coverage (van Geffen et al., 2021). The pixel size of TROPOMI was initially $3.5 \times 7$ km$^2$ and was reduced to $3.5 \times 5.6$ km$^2$ (August 2019), while the swath width is 2,600 km. The imaging spectrometer of this instrument measures radiation in the UV, VIS, near infrared, and shortwave infrared spectral regions (De Smedt et al. 2021; Veefkind et al., 2012). Operational level 2 (L2) products retrieved from TROPOMI observations include vertical columns of $O_3$, $NO_2$, HCHO, carbon monoxide (CO), sulfur dioxide ($SO_2$), and methane ($CH_4$). In the following analysis, we used the official TROPOMI $NO_2$ and HCHO products from the Tropospheric Emission Monitoring Internet Service and the Copernicus Open Access Hub, respectively, recorded between January 2019 and December 2020. The datasets were binned and averaged over a regular grid of $0.1 \times 0.1°$ to perform various statistical analyses at each location (i.e., grid box).

The TM5-MP-DOMINO $NO_2$ data used in this study are based on the DOMINO retrieval (van Geffen et al., 2019), previously used for Ozone Monitoring Instrument (OMI) data, and exploit the 405–465 nm spectral range to render tropospheric vertical column density with precision of 30–40%. The profile shape of the TM5-MP model is used for computation of the air mass factor (AMF) without accounting for aerosols. Screening of TROPOMI $NO_2$ data involved retaining data with a quality flag (QF) value higher than 0.5 and a cloud fraction (CF) lower than 0.2. The TM5-MP-DOMINO $NO_2$ dataset combines the versions 1.2.x. and 1.3.x. Version 1.3.x. was introduced on 2019/02/06, so, essentially, it covers the entire period here examined. Minor differences exist between the versions 1.2.x. and 1.3.x. and, according to all past studies, we combined them (Van Geffen et al., 2021). These differences were caused by improvements in the FRESCO-S algorithm devoted to retrieving cloud information. Indeed, since version 1.3.x., to avoid non-physical cloud fraction and pressure values, when the top of atmosphere reflectance is lower than expected, the surface albedo is reduced to match the top-of-atmosphere reflectance (Van Geffen et al., 2021). A further change in FRESCO is the treatment of very high cloud fractions (Van Geffen et al., 2021). Nevertheless, we excluded observations retrieved under these conditions.

A retrieval algorithm developed for OMI QA4ECV products is used to retrieve TROPOMI L2 HCHO vertical column densities. HCHO slant column densities are retrieved in the 328.5–359 nm fitting window. The HCHO vertical columns are calculated from the retrieved slant column densities and AMF, which is based on TM5-MP. TROPOMI L2 cloud products provide cloud information for AMF calculations. A more explicit explanation of the TROPOMI HCHO retrievals and their validation is provided in the works of De Smedt et al. (2018, 2021). Data filtering was performed as detailed in the product

read-me file (http://www.tropomi.eu/sites/default/files/files/publicSentinel-5P-Formaldehyde-Readme.pdf). The current
TROPOMI HCHO product is based on version 2.1.3 after 2020/07/13 and version 1.1.x for the period before (De Smedt et al., 2021). The version 2.1.3 includes various improvements compared to the previous version 1.1.x., such as a new surface albedo retrieval algorithm, the adoption of new OCRA cloud-free maps, and the correction of some QF values over snow/ice regions (the latter did not affect the investigated area). More information can be found in the product read-me file and recent validation activities (De Smedt et al., 2021).

**2.1.3 OMI**

OMI is onboard the National Aeronautics and Space Administration (NASA) Aura satellite, which has a Sun-synchronous polar orbit. OMI passes the Equator around 13:40 LT (local time) and the size of its pixels is 13 km × 24 km at nadir.

We used OMI $NO_2$ Level-2 data (over Chiba University and Tsukuba sites) and the recently updated Level-3 daily global gridded (0.25 × 0.25°) OMNO2d data V4. The algorithm includes improved surface and cloud treatments (Lamsal et al.,
2021). In the following analysis, we exploited observations recorded in 2005–2020, and screened out $NO_2$ data associated with a cloud fraction larger than 0.3. OMI and TROPOMI $NO_2$ datasets are based on slightly different retrieval algorithms, including cloud algorithms to estimate cloud fraction. Moreover, the size of the field of view of OMI is larger than that of TROPOMI, and OMI is affected by the row anomaly problem (http://omi.fmi.fi/anomaly.html), which further reduces the number of available observations. Here, the slightly larger cloud fraction threshold for OMI than that of TROPOMI tends to
counteract the smaller number of available OMI observations (by adding a further 10% of data. More information on the variation of OMI $NO_2$ as a result of modifying screening criteria can be found in Compernolle et al. (2020)). Since OMI $O_3$ only has some low sensitivity to the boundary layer while TROPOMI $O_3$ is currently limited to tropical latitudes, we did not use satellite-based $O_3$ datasets in this study.

**2.1.4 Surface in situ and additional ground observations**

In Japan, continuous monitoring of $NO_x$ and other compounds through a capillary surface network is performed by the Atmospheric Environmental Regional Observation System (AEROS). Historical time-series data are accessible to the public via the geographic information system (GIS) of the National Institute for Environmental Studies (NIES) (http://www.nies.go.jp/igreen/index.html). As the most recent years are unavailable from the NIES catalog, we limited our analysis to $NO_2$ observations in 2015–2018. We used a total of 266 stations, including general atmosphere measurement
stations and automobile exhaust gas measurement stations in Tokyo, Chiba and Ibaraki prefectures. The measurement principles of $O_3$, $NO_x$, and non-methane hydrocarbons are based on UV absorption, $NO$-$O_3$ chemiluminescence, and flame ionization detection, respectively.

In addition to the NIES dataset, we used observations of black carbon (BC) concentrations recorded by the continuous soot-monitoring system (COSMOS) (Kondo et al., 2009; Damiani et al., 2021) deployed at Chiba University site in November
2019. Our analysis focused on observations recorded in 2020.

Finally, we combined optical property information retrieved by a sky radiometer with MAX-DOAS data to estimate light-absorbing aerosols within the boundary layer (Damiani et al., 2021), as described in Section 2.2.

### 2.1.5 CLASS model

Simulations with a box model accounting for boundary layer dynamics and chemistry were performed using the Chemistry Land-surface Atmosphere Soil Slab (CLASS) model (van Stratum et al., 2012; Vilà-Guerau de Arellano et al., 2015). In these simulations, we used fixed dynamic (i.e., the usual boundary layer height recorded by the collocated lidar instrument) and chemistry conditions (as in van Stratum et al., 2012), along with the concentrations of $NO_2$ and HCHO recorded by the MAX-DOAS system within the boundary layer (at altitudes lower than 1 km) as initial conditions. The model solves the diurnal evolution of dynamical and chemical species over time in a well-mixed convective planetary boundary layer (PBL) (Vilà-Guerau de Arellano et al., 2015). Since these variables are assumed to be constant with height, the model output approximates MAX-DOAS observations in the PBL. Chemistry is represented by an $O_x$-$NO_x$-VOC-$HO_x$ photochemistry scheme based on 28 reactions that control $O_3$ formation. Although this simplified scheme omits other important organic species and aerosols, CLASS has been shown to reproduce the observed diurnal variability and mixing ratios of the main reactants present in polluted environments (Zara et al., 2021 and references therein).

### 2.1.6 Copernicus Atmosphere Monitoring Service (CAMS) global reanalysis

CAMS global reanalysis (EAC4) is the latest global reanalysis dataset of atmospheric composition produced by the European Centre for Medium-Range Weather Forecasts (ECMWF), including aerosols, chemical species and greenhouse gases (Innesset al., 2019). Atmospheric data are interpolated to 25 pressure levels (top level, 0.1 hPa) at a spatial resolution of $0.7 \times 0.7°$. Notably, both OMI (overpass around 13:40 LT) and GOME-2 (overpass around10 LT) $NO_2$ data are assimilated in CAMS, but anomalous emissions that occurred in 2020 are not included i.e., the simulations are based on a business-as-usual emissions scenario. Instead, CAMS uses MACCity anthropogenic emissions and the CO emissions upgrade described by Stein et al. (2014). Monthly mean VOC emissions were calculated using the MEGAN model.

### 2.1.7 Additional datasets

Ozonesonde observations were launched from the Tateno Aerological Observatory (Tsukuba, Japan) by the Japan Meteorological Agency (JMA), usually at 3 pm (i.e., around the anticipated time indicating maximum ozone). The KC ozonesonde used until November 2009 was replaced with an ECC ozonesonde in December 2009. In some of the following analyses, to fill the gap of the absence of MAX-DOAS $O_3$ observations in winter (see Sect. 2.1.1), we used data recorded in the period of 2013–2020 for consistency with the MAX-DOAS time series.

MERRA-2 is NASA's latest reanalysis and includes online aerosol fields that interact with model radiation fields (Buchard et al., 2017; Randles et al., 2017). This product is based on coupling of the Goddard Earth Observing System, Version 5

Earth system model with the Goddard Chemistry Aerosol Radiation and Transport aerosol model. The resolution is 0.625 x 0.5 degree with 72 vertical layers from the surface to approximately 80 km. In the following analysis, we employed data for wind speed, temperature, and PBL height (PBLH).

Following previous studies (e.g., Guevara et al., 2021), we used Google mobility data as a proxy for traffic counts as they are easily accessible for the majority of the countries and allowed us to compare the changes that occurred in different regions. Google Mobility data show changes to visits to various places worldwide (https://www.google.com/covid19/mobility/). Visits on each day are compared to baseline values for that day of the week. The baseline is the median value for the corresponding day of the week during the 5-week period of Jan 3–Feb 6, 2020. The following place categories are available:

Grocery & pharmacy, Parks, Transit stations, Retail & recreation, Residential, and Workplaces. Here, we focused on changes in the Transit stations category (Fig. 1) in certain large metropolitan areas of the world deeply affected by the pandemic. Google transit data has been previously used to estimate the emission reduction for the road transport sector (Guevara et al., 2021). It assumes that mobility trends in public transport hubs can be taken as a proxy for trends in road traffic emissions. This assumption is likely more appropriate for lighter vehicles than for heavier vehicles (Brancher, 2021).

## 2.2 Methods

Except where otherwise noted, we focus on ground-based daily observations recorded between 9 am and 3 pm LT. Weekly changes in $NO_2$, HCHO, $O_3$, and light-absorbing aerosols are reported as differences with respect the average value on weekdays. As the strongest reduction of anthropogenic emissions occurs on Sunday in the investigated region, we refer to weekly changes as the difference between the Sunday value and the average of the weekdays. The holiday effect was

260 estimated to be the difference between the average concentration of a given compound during the given holiday period and that in the two 10-day periods immediately before and after the holiday.

Light-absorbing aerosols within the boundary layer were estimated by combining sky radiometer and MAX-DOAS optical property data at UV wavelengths (Damiani et al., 2021). Then, in the following analysis, we examine the fine-mode absorbing AOD within the partial column below 1 km (i.e., fAAOD [0–1 km]), which was computed by combining the

265 columnar fine-mode fraction (FMF) and single scattering albedo (SSA) parameters retrieved from the sky radiometer with the mean partial column AOD estimated from MAX-DOAS measurements. As mineral dust can be strongly absorbing in the UV spectral range, we further removed days with Angstrom exponent (AE) < 1.

Under days characterized by stagnant low wind speed conditions, $NO_2$ accumulates around source locations. In contrast, under days with high wind speed conditions, $NO_2$ is dispersed. Tokyo is located in a polluted background with various

significant $NO_x$ sources surrounding it within about a 100 km radius. Therefore, due to the influence of surrounding sources, the outflow plume of $NO_2$ from Tokyo is not evident in the TROPOMI $NO_2$ maps. However, the spatial pattern of the difference between these two $NO_2$ composites, built based on wind speed data, reveals outflow patterns more clearly (see also Liu et al., 2016). We applied this method limitedly to Fig. 5c. To select the threshold values to identify high and low wind speed days for each pixel, we used MERRA-2 wind fields. According to previous studies (e.g., Fioletov et al., 2022), we

used a PBL averaged wind. Still, the results are not sensitive to the wind altitude because the wind is relatively constant within the boundary layer. Composite differences between high and low wind speed days in TROPOMI $NO_2$ were computed based on MERRA-2 wind fields averaged around the overpass time (12–3 pm). The median wind speed of each pixel was assumed to be the threshold between the high and low wind values. We first regridded the MERRA-2 data to the resolution of TROPOMI; then, for each grid cell, we computed $NO_2$ as the difference between the composite values of days with high and low wind speed.

## 3 Results

### 3.1 Trends and seasonal changes

The spatial distribution of the TROPOMI $NO_2$ and HCHO column data in the Kanto region is shown in Fig. 3 for 2019 and 2020 on an annual basis (top panels) and during the state of emergency (bottom panels). The bulk of $NO_2$ is around Tokyo, which is the most densely populated area (Fig. 2), including along the main transportation routes, and extends toward the south, where various large power plants and industrial activities are located (left column in Fig. 3). Overall, on an annual basis, column $NO_2$ over the GTA was reduced by about 10% in 2020 compared to 2019 (Fig. 3m), with larger absolute reductions around Haneda and Narita international airports, while smaller changes occurred in areas characterized by the presence of multiple power plants (South of Chiba). On the other hand, during the state of emergency, TROPOMI shows the largest $NO_2$ reduction (20–40%) in the southern Tokyo area while limited reductions (about 10%) occurred around Chiba and Tsukuba (Fig. 3o). The area affected by the most significant $NO_2$ reduction coincides with the region characterized by the larger decrease in traffic counts (Takane et al., 2022). Assuming that traffic currently contributes about 40% of Japanese domestic $NO_x$ emissions (Kurokawa and Ohara, 2020), a drop in mobility by about 50% in April–May (Fig. 1) is consistent with the TROPOMI-based estimate of $NO_2$ changes related to COVID.

Aside from anthropogenic emissions, meteorological conditions contribute to determining the interannual variability of $NO_2$. Annual differences between 2020 and 2019 in major meteorological parameters such as wind speed, surface temperature and PBLH, which are expected to influence the $NO_2$ distribution, were quite limited (Fig. S1a–c). In contrast, during the state of emergency, the wind speed in 2020 was slightly higher than that of 2019 and potentially contributed to further reducing the $NO_2$ levels (Fig. S1d).

Based on satellite observations and model simulations, Cooper et al. (2022) estimated a significant overall decrease in surface $NO_2$ over more than 200 cities around the world in April 2020 compared with 2019. Among others, they reported $NO_2$ changes for various Japanese cities. Within the Kanto region, they showed reductions peaking at Yokohama (-69%), more minor changes at Saitama (-32%), and values roughly in between at Tokyo (-54%). Despite the inverse correlation between the lockdown Stringency Index and the $NO_2$, they found that changes in Japan were comparable or slightly lower than those for the European cities where lookdown restrictions were much more stringent. In agreement with our findings (Fig. S1d and Fig. S3), they showed that changes in Japan could have been favored by meteorology and long $NO_2$ trends.

Although the period examined by Cooper et al. (2022) only partially coincides with the Japanese state of emergency, Fig. 3o shows comparable reductions. Moreover, Fig. 3o reveals the complex pattern of these variations, characterized by an evident North-South gradient with the most significant (negative) changes in Southern Tokyo, further evolving toward zero changes in the Saitama prefecture. This highlights the necessity of coupling detailed analysis at a regional scale with a large-scale study when examining COVID-related impacts, particularly when focusing on areas dominated by several close megacities.

Despite the high spatial heterogeneity of HCHO concentrations due to its short lifetime, the spatial distribution of the TROPOMI HCHO column was estimated (Panel (e–h) in Fig. 3). While $NO_2$ variability is dominated by anthropogenic activities, HCHO arises from both anthropogenic and natural sources. The principal source of HCHO is the oxidation of methane, which provides a global ambient background (e.g., Surl et al., 2018). Then, over continental atmospheres, the main anthropogenic sources are vehicle exhaust emissions and industrial emissions, while the main natural sources are plants and biomass burning (Surl et al., 2018; Sun et al. 2021; Ghahremanloo et al., 2021). Although the general increasing gradient from the ocean toward the continent resembles the pattern of $NO_2$, the HCHO distribution is not well defined and does not align with the urbanized region. The somewhat higher HCHO concentration in 2020 (Fig. 3n), despite lower anthropogenic emissions, may be driven by small differences in temperature between the two years (Fig. S1b). However, the change in meteorological conditions and the application of cloud screening cause the amount of data collected under clear sky conditions to be slightly different each year. Then, also the distribution along the year of the data can be different. For example, due to the rise in the summertime HCHO concentration, if frequent clouds caused few TROPOMI observations collected in the summer of a given year, the mean annual concentration of such a year could be smaller than the mean of the other year characterized by more summer HCHO observations. This confounding factor complicated the interpretation of HCHO changes. Then, despite the lower temperature that occurred during the state of emergency (Fig. S1e), TROPOMI did not show evident differences in the HCHO pattern between the two years (Fig. 3p).

A summer maximum characterizes the observed seasonal cycle of the HCHO columns shown in Fig. 4g. This indicates that biogenic emissions dominate HCHO even within our urban region. Pieces of evidence in TROPOMI HCHO reductions as a consequence of the COVID-related mobility restrictions have been reported only for China (Ghahremanloo et al., 2021) while meteorology likely drove most of the HCHO variations in India (Levelt et al., 2022). However, even in Wuhan, while the reduction in $NO_2$ reached about 83%, the decrease in HCHO was only 11%. The recent study by Sun et al. (2021) showed that comparable HCHO reductions (i.e., 11%) were found in the Northern China Plain for locations with predominant declines in $NO_2$ columns and elevated anthropogenic NMVOC emissions. However, reductions were favored by meteorological conditions. Then, simulations showed that most of the HCHO decrease resulted from the reduced anthropogenic $NO_x$ emissions. Still, an additional reduction in anthropogenic NMVOC emissions of about 15% would be necessary to match the observations (Sun et al., 2021). Since mobility restrictions in Japan were less severe and more gradual than those established in China, we expect such minor HCHO variations hardly identifiable by using satellite observations.

The HCHO-to-$NO_2$ concentration ratio is an indicator of near-surface $O_3$ sensitivity (e.g., Martin et al. 2004). Traditionally, the ozone production regime is considered to be VOC-limited when this ratio is lower than 1, $NO_x$-limited when it is higher

than 2, while ozone is expected to be in the transition regime when the values are in the range 1–2 (Duncan et al., 2010; Ryan et al., 2020). Although several studies used this ratio to infer $O_3$ sensitivity to $NO_x$ and VOCs by using observations from satellite and ground-based instruments (Duncan et al., 2010; Jin et al., 2017; Schroeder et al., 2017; Irie et al., 2021), some limitations still exist. Assuming the transition region lies within the range 1–2 (Duncan et al., 2010) could not be valid

at global levels, and it could be necessary to compute it depending on the region (Schroeder et al., 2017). Moreover, the ratio has an altitude dependence (e.g., Jin et al., 2017; Schroeder et al., 2017). While seasonal variations and trends in the columnar $HCHO/NO_2$ ratio (i.e., based on satellite observations) generally match the ratio computed with in situ observations, magnitudes are often different due to different vertical distributions of HCHO and $NO_2$ (Ryan et al., 2020). Therefore, although $O_3$ sensitivity derived from satellite column data can differ somewhat from that based on in situ

observations (Schroeder et al. 2017), it nonetheless provides useful information and has been extensively studied in relation to COVID-19 (e.g., Ghahremanloo et al., 2021 among others). The $HCHO/NO_2$ ratios are shown in Fig. 3i–l. Overall, the ratio increased in 2020, indicating a shift toward more $NO_x$-limited conditions. This change is particularly evident for Tsukuba, where the ratio rose from 2.1 to 2.9, while limited variations occurred over Tokyo and Chiba. We can observe similar findings during the emergency period (Fig. 3k,l).

To further contextualize the changes that occurred in 2020, Fig. 4 shows monthly partial column $NO_2$, $O_3$, HCHO, as well as light-absorbing aerosols within the boundary layer (i.e., < 1 km) recorded at Chiba University. To better account for the spatial heterogeneity of tropospheric gases, the average values from four MAX-DOAS systems looking at different directions were employed (Section 2.1.1 and Fig. 2). As shown in a recent study (Irie et al., 2021), both $NO_2$ and HCHO

were reduced and $O_3$ increased slightly during the period of 2013–2019. Any potential COVID-related effects in 2020 were superimposed over these trends. Indeed, in 2020, $NO_2$ remained at its lowest recoded levels for almost the entire year, whereas HCHO was only occasionally lower in 2020 than 2019, particularly in the second half of the year, and no modulation of $O_3$ was evident. When analysis was limited to the period of the state of emergency (i.e., roughly April-May), all species considered here showed decreases in May compared with the same period of 2019 while some enhancements in

April. This is coherent with an electricity demand reduction in May more significant than April for Chiba and the other Prefectures of the Kanto region (data from the Japanese Agency for Natural Resources and Energy available at https://www.enecho.meti.go.jp/statistics/electric_power/ep002/results_archive.html).

Overall, in agreement with the tropospheric column observations (Fig. 3o,p), slight changes occurred in the boundary layer around Chiba during the emergency period. In addition to the decrease in $NO_2$, similar month-to-month variabilities in

HCHO and $O_3$ are apparent from the differences between 2020 and 2019.

Miyazaki et al. (2021) showed that the pandemic caused a reduction in global NOx emissions which resulted in an overall decreased free-tropospheric ozone and some isolated enhancements, due to titration effect, at the surface in correspondence with some strongly urbanized regions (mostly in China). Although they mostly focused on a global scale, so their study is hardly comparable with our findings at regional scale, this expected opposite change of ozone with the altitude suggests that

MAX-DOAS ozone could results in negligible changes due to summing up of positive and negative changes within the column.

As a further reference, we compared the observed changes with equivalent CAMS data (Innesset al., 2019), which assimilate satellite observations of tropospheric $NO_2$ (Section 2.1.6). CAMS data roughly reproduced the observed interannual and seasonal variabilities (Fig. 4). However, while the trends in $NO_2$ and $O_3$ were comparable to observations, CAMS did not reproduce the HCHO decrease; instead, an increasing trend was simulated. The month-to-month variability was very similar to that of observations. Nevertheless, the CAMS $NO_2$ difference between 2020 and 2019 was generally smaller than the observed difference (Fig. 4i). Although satellite $NO_2$ data are assimilated in CAMS, the impact of assimilation is expected to be limited for short-lived species such as $NO_2$ (Inness et al., 2019). Therefore, as CAMS simulations did not include the anomalous 2020 emissions, comparison of CAMS and MAX-DOAS datasets supports the possibility that the emissions reduction in 2020 was responsible for the observed stronger $NO_2$ decrease. On the other hand, CAMS satisfactorily reproduced the month-to-month variabilities of both $O_3$ and HCHO.

Finally, we considered the changes in light-absorbing aerosol in both the boundary layer and the total column. Although this dataset is characterized by high uncertainty (Damiani et al., 2021), the values in 2020 were clearly the lowest on record. The largest relative changes (not shown) occurred before the state of emergency (i.e., in January and February), while the largest absolute change occurred in fall; nonetheless, some reduction was apparent in May. In this huge urbanized region, light-absorbing aerosols tend to be produced mainly by local pollution. However, transboundary transport has been shown to further modulate the light-absorbing aerosol dataset (Damiani et al., 2021), complicating the attribution of effects to the pandemic.

## 3.2 Weekend, holiday, and wind effects from various platforms

Previous studies (e.g., Beirle et al., 2003) have reported that apparent signatures of anthropogenic activity reflect a weekly cycle of $NO_2$ over most major cities in the northern hemisphere. To contextualize the weekly changes occurring in the GTA, we analyzed global OMI $NO_2$ data for the period of 2005–2020. Figure 5a and Table S1 show the global ranking of the resulting relative changes on Sunday over cities with population larger than 0.5 million inhabitants (we implicitly excluded cities where the rest day is on Friday; see Stavrakou et al., 2020). For the sake of clarity, we report only those cities with changes more prominent than 30%, although the majority of locations showed negative values (except in China; Stavrakou et al., 2020). Japanese cities dominate the histogram; worldwide, the extent of changes over the GTA is exceeded only by Los Angeles and São Paulo. The OMI-based map of such changes (Fig. 5b) shows a well-defined pattern over Japan, with consistent values negative across the country, which were lower than –40% over most of the GTA and industrialized regions in southern Japan (i.e., Nagoya, Osaka).

Power plants and other industrial facilities are the dominant stationary emissions sources, and are generally consistent, with little effect of weekly modulation. Satellite $NO_2$ data can be used to investigate both stationary (Beirle et al., 2019) and

mobility-related emissions sources (Ialongo et al., 2020). Using satellite $NO_2$ and wind field data, emissions related to large and isolated stationary sources can be assessed (Beirle et al., 2019). Nevertheless, when multiple stationary sources are

located within an urbanized region, as in our study area, evaluating the impacts of the various sources becomes challenging. As shown in Fig. 5c, we exploited the high resolution of TROPOMI data to compare $NO_2$ changes associated with the weekly cycle (i.e., Sunday minus weekdays, red contours) to $NO_2$ changes associated with wind speed (high wind speed days minus low wind speed days, black contours; see Sect 2.2 for further details) in 2019–2020. Moreover, to better highlight the spatial patterns of these changes, we focused on the extended summer period from April to September, when the $NO_2$

lifetime is shorter and $NO_2$ tracks emissions sources better. Both contour lines suggest larger reductions over Tokyo than other areas, with patterns mostly matching the spatial distribution of population density. The weekly related $NO_2$ generally extends north to Tsukuba and east to Chiba with a roughly homogeneous spatial gradient, while the gradient sharpens south of Chiba.

During days characterized by stagnant wind conditions, $NO_2$ levels tend to accumulate around stationary sources, while rapid

transport occurs with stronger winds. In addition to covering areas with high population density, high negative wind-related changes are apparent around the main power plants on both sides of Tokyo Bay. Moreover, detailed distributions around stationary sources have been revealed (e.g., changes around Narita international airport and the isolated Kashima power plant on the east coast of Chiba Peninsula). Overall, the distribution of wind-related changes is more southerly than the bulk of weekly changes. Moreover, this gradient becomes positive in the north, highlighting the region downwind of the Tokyo

area, including Tsukuba. Overall, wind-related $NO_2$ changes closely resemble the pattern shown in the Japanese Emission Inventory.

The map in Fig. 5d shows the $NO_2$ decrease occurring during the end-of-year holiday, when most anthropogenic emissions are reduced. To quantify this holiday effect, the average $NO_2$ value of the 10 days before and after the holiday (defined here as the period from 25 December to 4 January) was subtracted from the holiday mean. Figure 5d shows similar patterns to

Fig. 5b, although the longer $NO_x$ lifetime in winter causes the changes to be less confined to urbanized regions. TROPOMI $NO_2$ decreased by about –43 and –49% in Chiba and Tsukuba, respectively (Fig. S2). The holiday effect is apparent in almost all cities with intense weekly cycles (Fig. 5a) and will be used as a further reference.

Multi-year MAX-DOAS observations of partial column $NO_2$ recorded at Chiba University also showed a decrease around the end of the year (Fig. S2). It was about –44%, while smaller changes occurred in HCHO (about –15%). Slightly larger

reductions were observed at Tsukuba site (–55% and 29%, for $NO_2$ and HCHO, respectively).

Focusing on the holiday period provides insights into the response of $O_3$ to these significant changes. As only data at SZA lower than 50° can be used to retrieve $O_3$ with the DOAS method (Irie et al. 2011; see Sect. 2), we focused on ozonesonde profiles recorded at the Tateno Observatory, located about 50 km from Chiba. In this manner, we assessed the vertical distribution of $O_3$ changes within and slightly above the PBL. These data were used to better interpret the changes in partial

column $O_3$ estimated by the MAX-DOAS systems. In the lower troposphere, the average $O_3$ profile concentration averaged across business days was smaller than the $O_3$ concentration recorded during the end-of-year holiday (Fig. 5e). The difference

was greatest in the lowest layers (about –18%) and remained nearly constant up to 0.5 km; then, it became smaller and disappeared above 1 km. The reduced NO titration effect occurring under VOC-limited conditions is likely the main driver of this peculiar pattern of increased $O_3$ levels in the lowest layers during the holiday period.

We simulated the periods of 2019 and 2020 with the CLASS model using the appropriate mean $NO_2$ and HCHO observations and lidar-based PBLH recorded at Chiba as initial conditions. The results of the simulations (vertical lines in Fig. 5e) confirmed the holiday-associated positive $O_3$ enhancement, with a relative difference of 22% at 15:00 LT, which was also the time of the sonde launch corresponding to the diurnal ozone peak. Previous studies have reported comparable $O_3$ enhancement effects on the weekend, particularly in winter (Sadanaga et al., 2012; Sicard et al., 2020a). Using the same

approach, we evaluated the changes occurring around mid-August, which is a period of reduced mobility. Such changes are interesting in terms of the $O_3$ formation regime, but significant variability prevented clear identification of trends (Fig. S2).

Figure 6 further characterizes the $NO_2$ weekend effect and contextualizes the holiday effect through comparison of ground-based partial column (< 1 km), satellite-based tropospheric column, and surface in situ observations. In Chiba, relocation of

all four instruments occurred in 2014 (i.e., the pointing direction of some sensors was changed slightly; Irie et al., 2021); therefore, in the following analysis, we excluded observations recorded in 2013 and 2014. Panel (a) shows the geographic distribution of $NO_2$ weekly changes on Sunday for OMI, with MAX-DOAS and in situ observations plotted together. We focused on 2015–2018, when all three datasets were available. The spatial distribution of OMI $NO_2$ is as shown in Fig. 5b. The largest negative changes occurred over Tokyo, which reached –45%, and changes became smaller toward Chiba and

Tsukuba. Notably, the magnitude of these changes is larger around Tsukuba (–35%) than around Chiba (–30%), although Tsukuba is a suburban site, while the urban area of Chiba hosts a larger population (Fig. 2) and industrial activities. In situ observations match the satellite-based spatial distribution, although with a somewhat smaller magnitude (by about 5%). MAX-DOAS partial column data reproduced the spatial distribution of the previous datasets (inset in Fig. 6a), showing a magnitude of change slightly closer to that of OMI (i.e., total column) than the surface network. Both in situ and MAX-

DOAS observations were averaged over 9 am–3 pm, while OMI observations were generally recorded around 1:30 pm. The small incongruence among measurements is likely due to the different sampling periods (i.e., while in situ observations include data recorded under all meteorological conditions, satellite and MAX-DOAS data were limited to clear-sky days). We also compared the climatology of the weekly cycle from various platforms according to data availability (excluding 2020). For comparison with the other datasets, we used the mean value from the in situ stations within Chiba (Fig. 6b) and

Ibaraki (Fig. 6c) prefectures, respectively. Despite the different periods analyzed, the weekly cycles were very similar for all three platforms. The magnitude of change is largest on Sunday; at Chiba, OMI changes are approximately 10% larger than both ground observation types. By contrast, at Tsukuba, in situ data are lower than OMI and MAX-DOAS by 10–15% (but due to the scarcity of in situ stations around Tsukuba, the average value of stations inside Ibaraki prefecture is likely not representative of the area around Tsukuba). In addition to Sunday, indications of reduced $NO_2$ are apparent on Friday and

Saturday at both sites. As a further reference, we included the $NO_2$ weekly cycle of Tokyo Prefecture. The data confirmed

that weekly cycles are similar in the three prefectures, with values for Tokyo on Sundays being about 5–10% larger than in other prefectures.

Figure 6d and 6e show changes in OMI $NO_2$ on Sunday averaged over the periods of 2005–2007, 2015–2019, and 2005–2019 at Chiba and Tsukuba, respectively. The OMI $NO_2$ column over the investigated region decreased by about 50% over 2005–2019 (see Fig. S3 for Chiba). Therefore, in accordance with a previous study (Stavrakou et al., 2020), the weekly cycle showed a reduced amplitude in recent years. During this latest period, in situ, satellite, and MAX-DOAS observations at Chiba coincided, while these data sources showed a spread of about 15% at Tsukuba. This larger variability at Tsukuba is likely due to its lower $NO_2$ levels and suburban location downwind of the Tokyo area (Fig. 5c).

As a further reference, Fig. 6d,e also show the end-of-year holiday reductions. Overall, when considering the full 2005–2019 period, OMI holiday changes were larger than OMI Sunday changes by about 8% for both Chiba and Tsukuba. However, in recent years, $NO_2$ values were much larger on holidays than on Sundays.

For Chiba, the distinct Sunday changes recorded by each of the four MAX-DOAS instruments were also investigated. The instruments showed a spread of less than 10% with limited interannual variability. The amplitude of the $NO_2$ weekend effect differed among instruments, with smaller effects for the instruments pointing south and east than for those pointing north and west. This difference was likely due to the presence of power plants and industrial activities within a few tens of km south of the sampling site, providing an additional and more constant emissions source that reduces the difference between rest and business days. Indeed, absolute $NO_2$ amounts were generally higher for the MAX-DOAS facing south, followed by the east-, west-, and finally north-facing instruments (inset in Fig. S3).

Moreover, changes in MAX-DOAS $NO_2$ were much more prominent in 2020 than in previous years. Notably, 2020 was equally anomalous for all instruments, with decreases around 15–20%. Therefore, the observed decrease was not a local phenomenon. Generally, MAX-DOAS changes at Tsukuba are more significant than the data recorded at Chiba. Nevertheless, at Tsukuba, changes in 2020 were within the usual level of interannual variability. Therefore, the anomalous weekly cycle in 2020 affected the urban region of Chiba but not suburban area.

### 3.3 Weekly cycles

### 3.3.1 Weekly cycle of partial column $NO_2$, HCHO, $O_3$ and light-absorbing aerosols

Figure 7 shows the interannual weekly changes in MAX-DOAS partial columns of $NO_2$, HCHO, $O_3$, and HCHO/$NO_2$ at Chiba and Tsukuba sites. Further changes in fAAOD [0–1 km] were reported for Chiba only. As we used observations averaged over four independent MAX-DOAS systems at Chiba, the data are more statistically robust and representative of a larger area than observations from Tsukuba (sampled with only one instrument). Therefore, most of the following discussion on interannual changes is focused on Chiba.

As noted above, at Chiba, while the $NO_2$ change on Sunday was approximately –30% for 2015–2019, it reached about –45% in 2020. This difference was much larger than 2 × sigma (Fig. S4). On the other hand, except in 2020, interannual variability

was much smaller on Sunday than on other days. By contrast, at Tsukuba, Sunday changes in 2020 were comparable to those in previous years.

Usually, the weekend effect was absent in HCHO data from both Chiba and Tsukuba. Nevertheless, at Chiba, negative HCHO changes on Sunday 2020 were unprecedentedly large (around –30%). This negative anomaly is larger than the largest deviation on any other day in previous years, and can be attributed to abnormally low anthropogenic NMVOC on the Sundays of 2020 (Sun et al., 2021; Ghahremanloo et al., 2021). By contrast, HCHO showed no unusual trends at Tsukuba in 2020, and this difference is likely due to differences in local conditions (i.e., Tsukuba is located in a suburban area, where a

greater amount of biogenic VOC emissions contribute to HCHO concentrations) and higher variability associated with the usage of only one instrument.

A weekly cycle in the $HCHO/NO_2$ ratio is evident for Tsukuba, with its peak on Sunday (approximately double the weekday average). On the other hand, for Chiba, the slightly higher ratio on Sunday was similar to the value on weekdays (0.55 vs. 0.4). Notably, on almost all days of 2020, the $HCHO/NO_2$ ratio at Tsukuba was higher than in previous years, while no

difference was recorded at Chiba. Overall, due to the seasonal variations in $NO_2$ and HCHO concentrations, the $HCHO/NO_2$ ratio also shows significant seasonality, with a large ratio in summer compared to the other seasons (Irie et al., 2021). Nevertheless, the frequent cloudy conditions in the late-sprint to summer period and the limited temporal extension of the dataset prevent to evaluate potential seasonal differences in its weekly cycle.

Despite this large variation in the main precursors, corresponding modulations in the O3 partial column were not recorded on

Sundays of 2020 or previous years. Although the amplitude of the weekend effect in ozone is likely to have been reduced in recent years (Sicard et al., 2020a), its absence in our data contrasts with previous results showing a discernible weekend effect in surface ozone at Tokyo (Sadanaga et al., 2012). Ozone in the free troposphere presents a greatly smoothed diurnal cycle, if any, compared to surface ozone and this diurnal cycle is generally strongest below 950 hPa (Petetin et al., 2016). The ozone profile is different from the other trace gases. In contrast to $NO_2$, which strongly decreases its concentration with

altitude, ozone concentration does not decrease with altitude. The ozone weekend effect at the surface level is usually 10% (Sadanaga et al., 2012) and is much less evident than $NO_2$. As suggested by ozonesonde observations (Fig. 5e), ozone changes due to titration maximize at the surface and tend to reduce shortly at h > 0.5 km. Since MAX-DOAS O3 partial column observations sample the 0-1 km layer, the effect tends to disappear in our data. Moreover, more titration is expected in winter, but MAX-DOAS O3 observations were unavailable this season (Sect. 2.1.1). Finally, the number of MAX-DOAS

daily ozone samples was generally smaller than the other trace gases. Therefore, any potential weekly cycle would be difficult to observe in our MAX-DOAS O3 partial column dataset.

We further evaluated potential ozone differences between Sundays and weekdays of 2015–2020 using the box model (not shown). As in the previous simulations, we used MAX-DOAS observations of $NO_2$ and HCHO and lidar-based PBLH as initial conditions and focused on the period most strongly affected by the pandemic (i.e., April–September). The simulated

ozone differences between Sundays and weekdays were slightly negative (i.e., Sunday O3 < weekday O3) and ranged from –4% in 2018 to –8% in 2020.

As shown in the bottom panel, we examined the weekly cycle of fAAOD partial column data at Chiba. Although such data are characterized by high interannual variability, similar to the results presented above, 2020 data were anomalous, characterized by negative changes of about 50% on Sunday and no variation on Saturday. Notably, consistent with the results for $NO_2$ (Fig. 6d), changes in fAAOD on Sundays in 2020 are comparable with the effect usually observed during the end-of-year holiday. Although the weekly cycle is hardly discernible in the other years, the weekly cycle in 2020 is coherent with that of collocated observations of surface BC mass concentration, which show reductions larger than 40% on Sunday and no change on Saturday.

We further excluded the influence of meteorology on the observed interannual variation by examining data on wind speed, wind direction, and temperature recorded at Chiba on days with available MAX-DOAS observations (Fig S5). Overall, the results showed that wind did not drive the weekly changes in tropospheric gases. As increasing temperature enhances biogenic emissions and boosts oxidation processes, temperature is usually positively correlated with HCHO. We verified the absence of apparent weekly variation in temperature. Moreover, due to the large amount of data recorded at Chiba site, we could confirm that weekly changes around the satellite overpass time were representative of the daily $NO_2$. No apparent difference between MAX-DOAS daily $NO_2$ for 9:00–15:00 and for 12:00–15:00 was observed (the same result was obtained for HCHO and $O_3$).

### 3.3.2 Weekly cycle of TROPOMI $NO_2$ tropospheric columns

Next, we examined the pattern of weekly changes in TROPOMI $NO_2$, looking for differences in the spatial distribution and magnitude of Sunday changes between 2019 and 2020 (Fig. 8). Overall, changes were larger in 2020 than in 2019, reaching –50% over central Tokyo. Moreover, during the extended summer period (April to September), when the largest influence of COVID is expected, differences between 2020 and 2019 were even more apparent. In addition to the magnitude of the differences, the area characterized by higher $NO_2$ concentrations was greatly reduced in 2020 compared to 2019. This reduction averaged about –27% for weekdays and reached –67% for Sundays (Fig. S6).

Within the areas sampled by the MAX-DOAS systems, TROPOMI showed no clear differences between 2019 and 2020 in either Chiba or Tsukuba. For Chiba, this contrasts with ground observations. Different time windows, over which the daily means were computed (i.e., between 9 am and 3 pm for MAX-DOAS and around 1.30 pm for TROPOMI), and cloud screening procedures resulted in a larger amount of MAX-DOAS data than satellite data and likely contributed to these differences. However, despite being based on one year of observations, both TROPOMI maps could correctly reproduce the spatial pattern of changes estimated from ground-based observations around Chiba, i.e., the north-south gradient driven by the presence of power plants. This similarity provides confidence in identifying the interannual variability of the spatial distribution of the $NO_2$ weekly cycle based on TROPOMI data.

## 4. Discussion and conclusions

This study investigated the interannual, seasonal, weekly variabilities and spatial distributions in $NO_2$, HCHO, $O_3$, and light-absorbing aerosols measured with multiple platforms within the Greater Tokyo Area, which is the most populous metropolitan area in the world. We mainly examined the period of 2013–2020, focusing on 2020, when an effect from COVID-19 is expected. The main results can be summarized as follows:

• In 2020, levels of $NO_2$ and light-absorbing aerosols were the lowest on record, but the potential COVID-19 impact was
superimposed on a decreasing trend.

• At Chiba, MAX-DOAS observations within the PBL showed that annual $NO_2$ reductions in 2020 were about 10% relative to 2019 with limited changes during the period of the state of emergency. No deviations in $O_3$ and HCHO were apparent.

• TROPOMI column-based observations confirmed the observed reduction in $NO_2$ and the absence of relevant changes in HCHO. During the state of emergency, $NO_2$ reductions exceeded 40% in the southern Tokyo area but were about 10% over
Chiba and Tsukuba. Moreover, both satellite and MAX-DOAS observations showed enhancement of the HCHO/$NO_2$ ratio, which was strongest in the suburban region.

• OMI observations demonstrated that the weekly and holiday effects in $NO_2$ within the GTA are among the largest in the world. $NO_2$ reductions on rest days are not limited to the GTA but extend uniformly over most of Japan.

• Surface in situ, MAX-DOAS partial column and satellite-based tropospheric column observations showed a coherent $NO_2$
weekly cycle, with the largest reductions on Sunday. Ground observations aligned with the spatial distribution of the satellite changes, even within the relatively limited area sampled by the MAX-DOAS systems.

• In 2020, ground and satellite observations showed an anomalous weekly cycle in $NO_2$ in urban areas, with larger reductions on Sunday than in previous years. Similar large changes in light-absorbing aerosols were identified. Such changes are comparable to the reductions observed during the end-of-year holiday period.

• At Chiba, large $NO_2$ reductions on Sunday were coupled with simultaneous reduction of HCHO, whereas no significant changes in $O_3$ were observed.

• In Japan, the reduction in mobility in 2020 was more extensive on the weekend than on business days, in accordance with the larger $NO_2$ weekly change in 2020 found in the urban areas. By contrast, other countries generally showed the opposite behavior. This highlights modification of habits by the Japanese populace that resulted in unique air quality effects,
suggesting widespread adoption of recommendations aimed at limiting the spread of the pandemic in Japan despite the lack of strict legal restrictions.

Although not explicitly mentioned in the previous discussion, an implicit assumption of our study relies on the fact that satellite observations available only around midday are representative of daily changes computed, for example, by hourly observations. Although we provide evidence that this is likely the case (see Fig. S5), data from new geostationary satellites
(e.g., Geostationary Environment Monitoring Spectrometer on board the Geostationary Korea Multi-Purpose Satellite 2) are expected to shed some further light on this issue.

A further shortcoming is the scarcity of reliable satellite-based tropospheric ozone datasets to complement the satellite-based spatial distribution achieved with $NO_2$ and HCHO observations. Despite the recent progress (Shen et al., 2019), OMI $O_3$ only has some low sensitivity to the boundary layer, and this would make challenging any analysis over the investigated region (past studies found some correlation with the actual surface ozone in China, where tropospheric ozone is much larger, Shen et al., 2019). TROPOMI is expected to improve this capability soon, but its ozone dataset is currently limited to tropical latitudes.

Finally, it is worth mentioning the potential impact of the rebound of the long-range transport of pollutants after the Chinese economic recovery from the COVID-19 pandemic (Itahashi et al., 2022) on the current pollution within the Kanto region will deserve further investigation.

**Data availability.** All datasets used in the present study are publicly available below:

https://www.temis.nl/

https://scihub.copernicus.eu/

https://disc.gsfc.nasa.gov/

https://woudc.org/

https://www.google.com/covid19/mobility/

http://atmos3.cr.chiba-u.jp/skynet/

http://atmos3.cr.chiba-u.jp/a-sky/

https://www.ecmwf.int/en/forecasts/dataset/cams-global-reanalysis

http://www.nies.go.jp/igreen/index.html

**Author contribution.** Conceptualization, A.D.; Methodology A.D. and R.C.C.; Measurements and data curation, H.I.; Analysis, A.D., D.B., S.H.H., and S.K.; Funding acquisition: H.I.; Writing—review and editing, A.D., H.I., D.B., R.R.C.; all authors have read and agreed to the published version of the manuscript.

**Competing interests.** The authors declare that they have no conflict of Interest.

**Acknowledgements.** This research was supported by the Environment Research and Technology Development Fund (JPMEERF20192001 and JPMEERF20215005) of the Environmental Restoration and Conservation Agency of Japan, JSPS KAKENHI (grant numbers JP19H04235, JP20H04320, JP22H03727, JP22H05004 and JP21K12227), the JAXA 2nd research announcement on the Earth Observations (grant number 19RT000351) and the Virtual Laboratory (VL) project by the Ministry of Education, Culture, Sports, Science and Technology (MEXT), Japan. R.R.C. acknowledges the support of the
Consejo Nacional de Ciencias y Tecnología (CONICYT, Preis ANILLO ACT210046). The authors would like to thank the OMI and TROPOMI Science Teams for the data products, CAMS and MERRA-2 for the corresponding reanalysis data products, and JMA, AEROS & NIES staff.

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

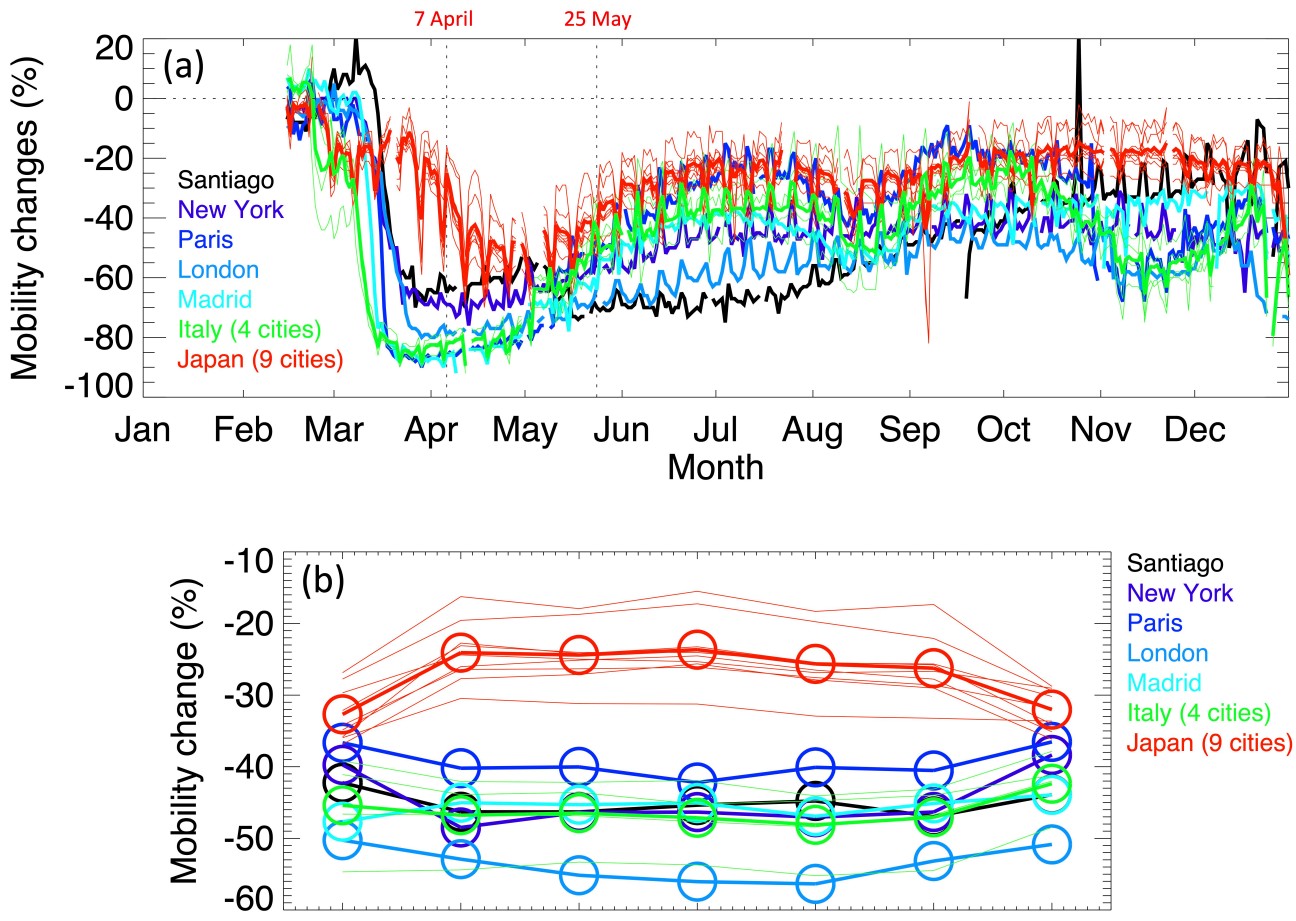

**Figure 1: Google mobility data for the place category Transit stations in 2020 (from February 15 to December 31) compared to the pre-COVID period, for selected large metropolitan areas around the world strongly affected by the pandemic. For Japan and Italy, values were averaged (thick line) over 9 (Tokyo, Kanagawa, Saitama, Chiba, Kyoto, Nara, Osaka, Fukuoka, Nagasaki) and 4 (Milan, Rome, Naples and Turin) cities/prefectures (thin lines), respectively: (a) time series; (b) weekly changes. Holidays were removed. The baseline in both panel (a) and panel (b) is the median value for the corresponding day of the week during the 5-week period of January 3–February 6, 2020.**

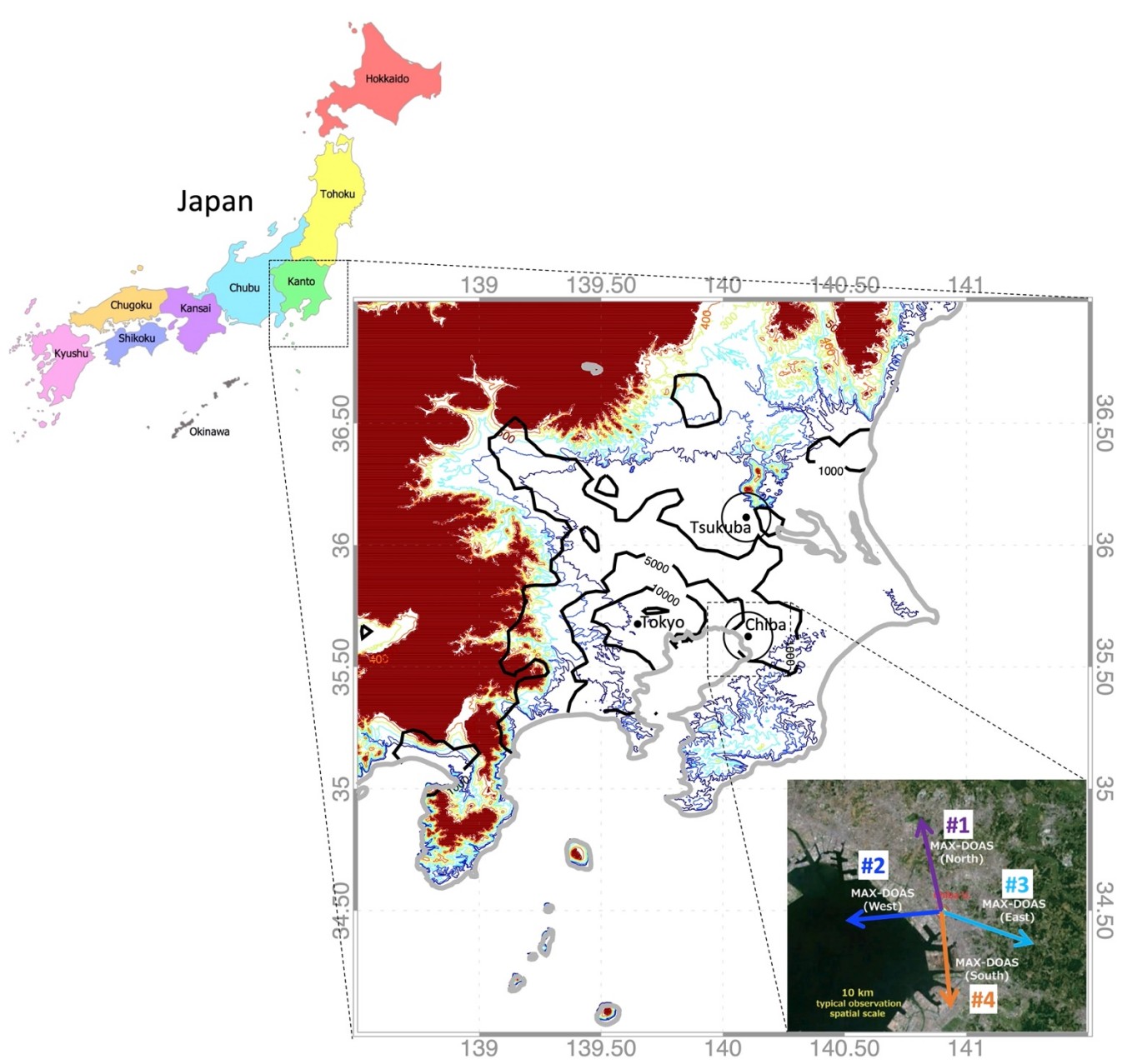

**Figure 2: Main panel: Terrain map (meters above sea level, colored isolines; elevations > 0.5 km shaded dark red) from the General Bathymetric Chart of the Oceans (GEBCO) 2021 and population density (black isolines, people/km2) from the Gridded Population of the World, Version 4 (GPWv4). Circles highlight the average area sampled by the MAX-DOAS system (four azimuthal directions for Chiba, one direction for Tsukuba). The top inset shows the location of the investigated Kanto region in Japan and the bottom inset (© Google Maps 2019) shows the azimuthal pointing directions of the four MAX-DOAS systems deployed at Chiba University.**

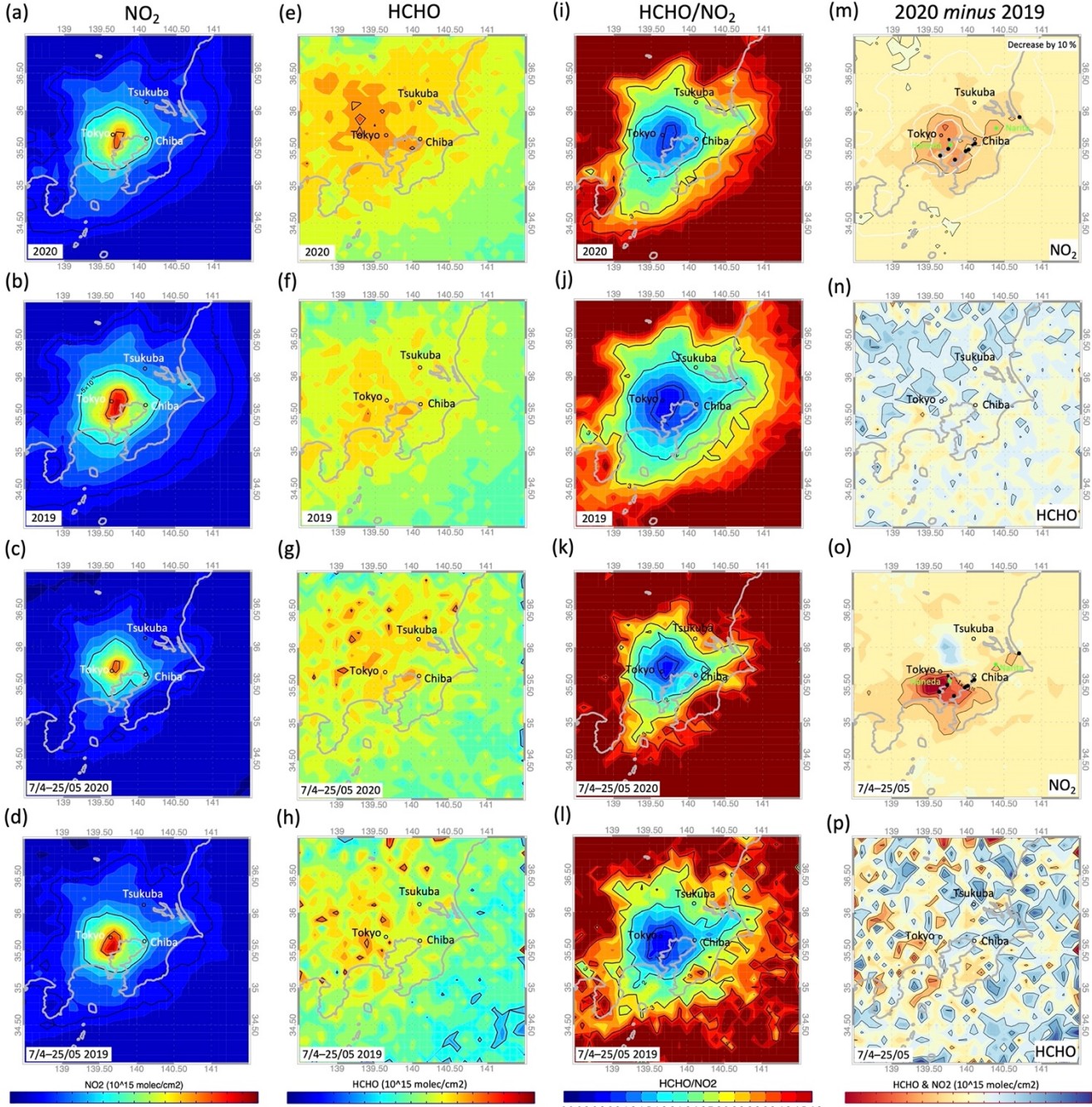

**Figure 3:** Top two panels: Spatial distribution of TROPOMI NO₂ (a,b), HCHO (e,f), and HCHO/NO₂ (i,j) in 2020 and 2019, as well as the 2020–2019 differences in NO₂ (m) and HCHO (n). Bottom two panels: Spatial distribution of TROPOMI observations is as described above but limited to 7 April–25 May (i.e., the state of emergency).

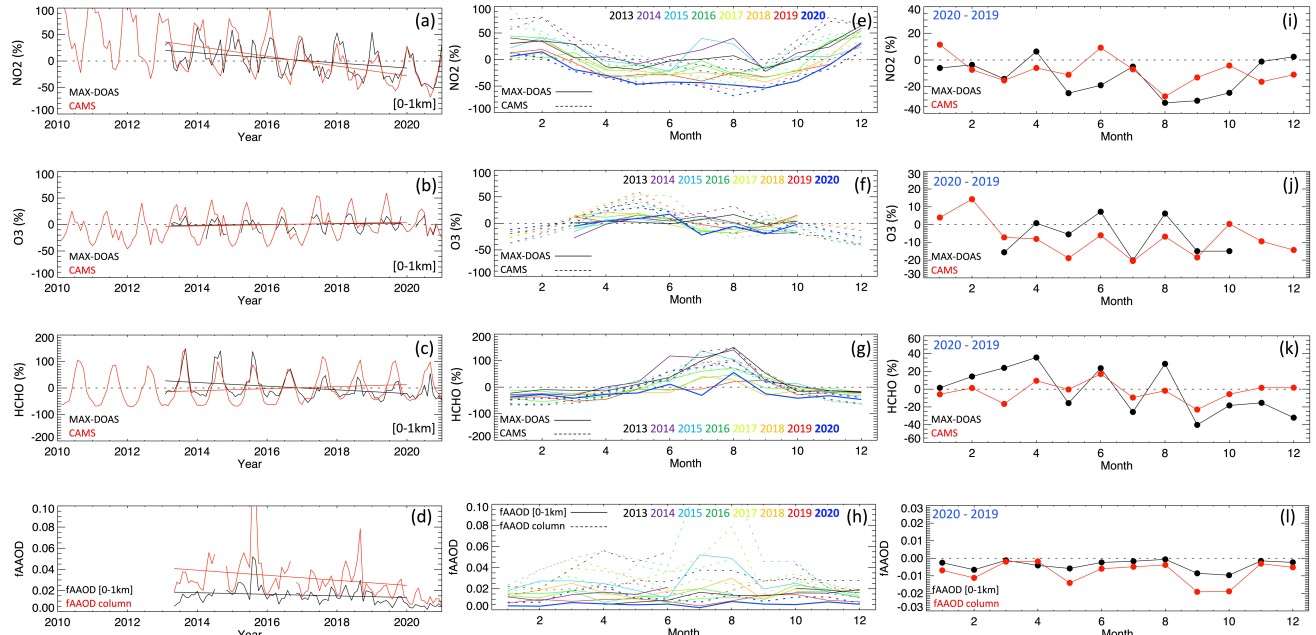


**Figure 4: Left column: monthly time series of NO$_2$ (a), O$_3$ (b), and HCHO (c) partial tropospheric column (< 1 km) as recorded by the MAX-DOAS system and estimated by CAMS at Chiba University. Central column: seasonal monthly changes in MAX-DOAS observations and CAMS estimates of NO$_2$ (e), O$_3$ (f) and HCHO (g). Right column: differences (2020 minus 2019) in MAX-DOAS**
**observations and CAMS estimates for NO$_2$ (i), O$_3$ (j) and HCHO (k). Results are shown as percentage changes with respect to the 2013–2019 average (left and central panels) and 2019 (right panel). Bottom panel: changes as described above but for fine mode light-absorbing aerosols i.e., fAAOD and fAAOD (0–1 km).**

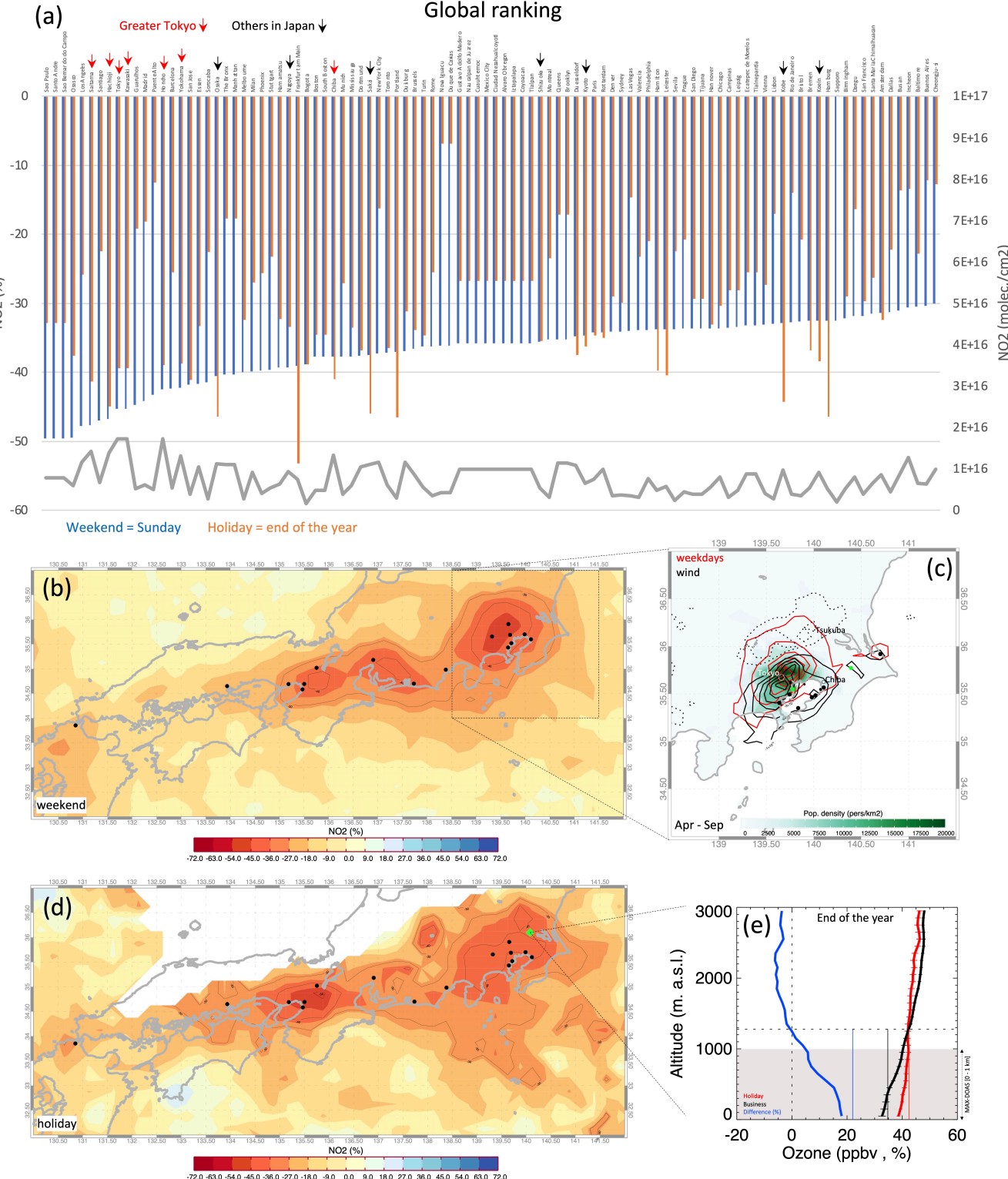

**Figure 5: Global ranking of the weekend effect (Sunday minus weekdays, blue) and holiday effect (end-of-year period minus business days, orange) for cities with a population greater than 0.5 million based on OMI $NO_2$ in 2005–2019. Only cities where changes were larger than 30% were shown. Opposite axis: mean $NO_2$ total tropospheric column in grey (a). The spatial distribution of the weekend effect over Japan ((b), relative changes) and over the Kanto region ((c), absolute changes limited to April–September. Red lines: Sunday minus weekdays; black points: power plants; green points: airports). Wind-related $NO_2$ changes limited to April–September (black lines: high minus low wind speed) and population density (filled contours, in green shades) are also plotted in (c). Contours for both red and black solid lines read as follows: -1,-2,-3,-4,-5,-6 $x10^{15}$ molec./cm$^2$. Then, the black dashed lines show positive changes (0.5, 1 $x10^{15}$ molec./cm$^2$). Spatial distribution of the holiday effect in $NO_2$ across Japan (d); the low number of OMI observations prevents determination of the difference along the coastline of the Japan Sea. Ozone profiles obtained from ozonesondes launched from Tateno (green point in (d)) during the end-of-year holiday period and business holidays in 2013–2020 (i.e., when MAX-DOAS data were available) are plotted along with box simulations (vertical lines) of the mean ozone concentration within the boundary layer (e). See Sect. 2 for further details.**

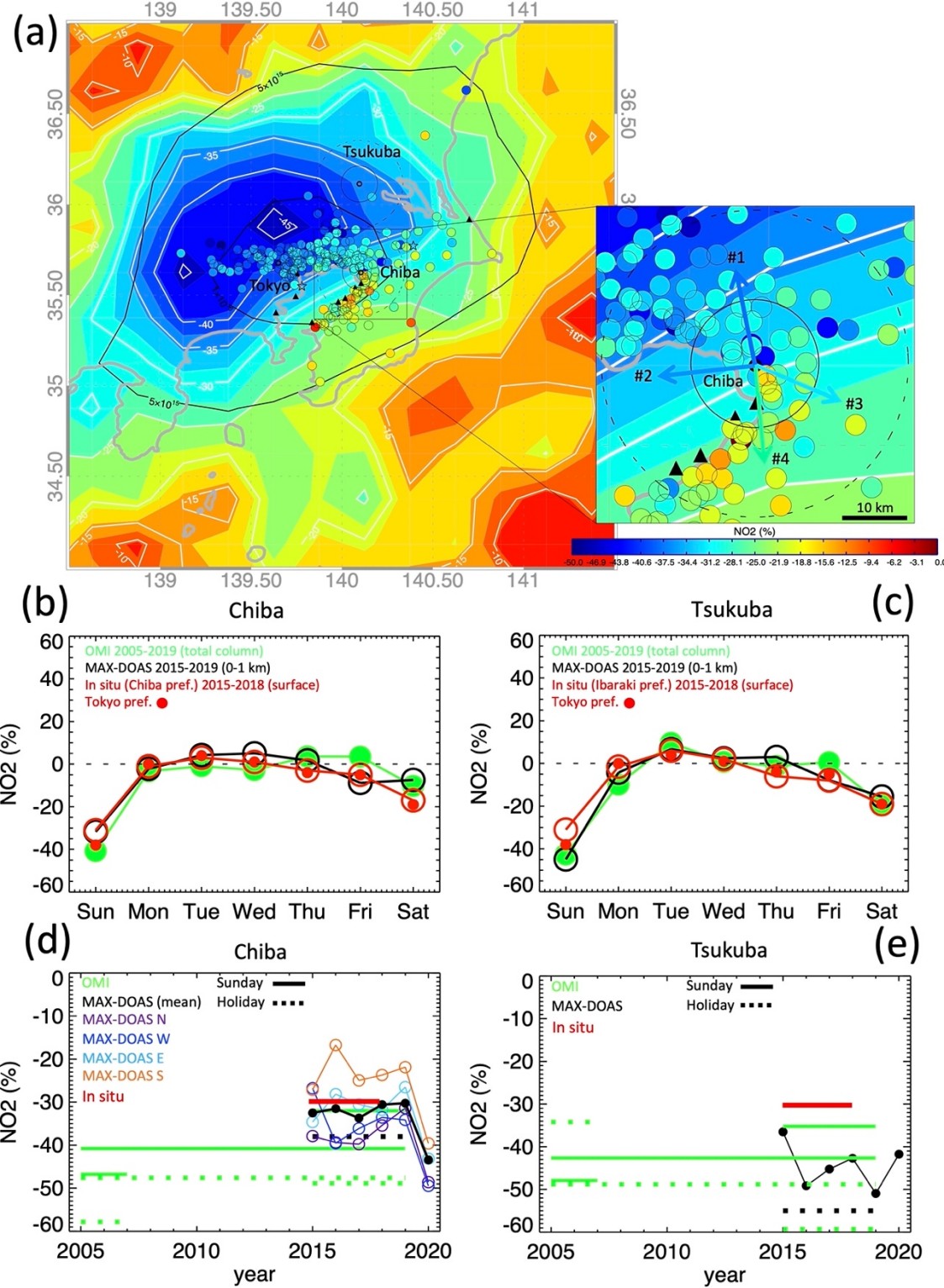

**Figure 6: Spatial distribution of the weekend effect (Sunday minus weekdays) based on OMI NO$_2$ total tropospheric column (filled contours), MAX-DOAS partial tropospheric column (< 1 km, colored arrows), and in situ NO2 observations (circles) over 2015–2018 ((a), main panel). Magnified view of the area around Chiba University (inset). NO$_2$ weekly cycle at Chiba and Tsukuba for the datasets noted above. In situ data were averaged across the prefecture (b,c). The weekend effect (on Sunday) and the holiday effect (at the end of the year) were averaged over different periods (as indicated by the horizontal lines) at Chiba and Tsukuba (d,e). MAX-DOAS NO$_2$ data are also plotted for each year and for each instrument at Chiba.**


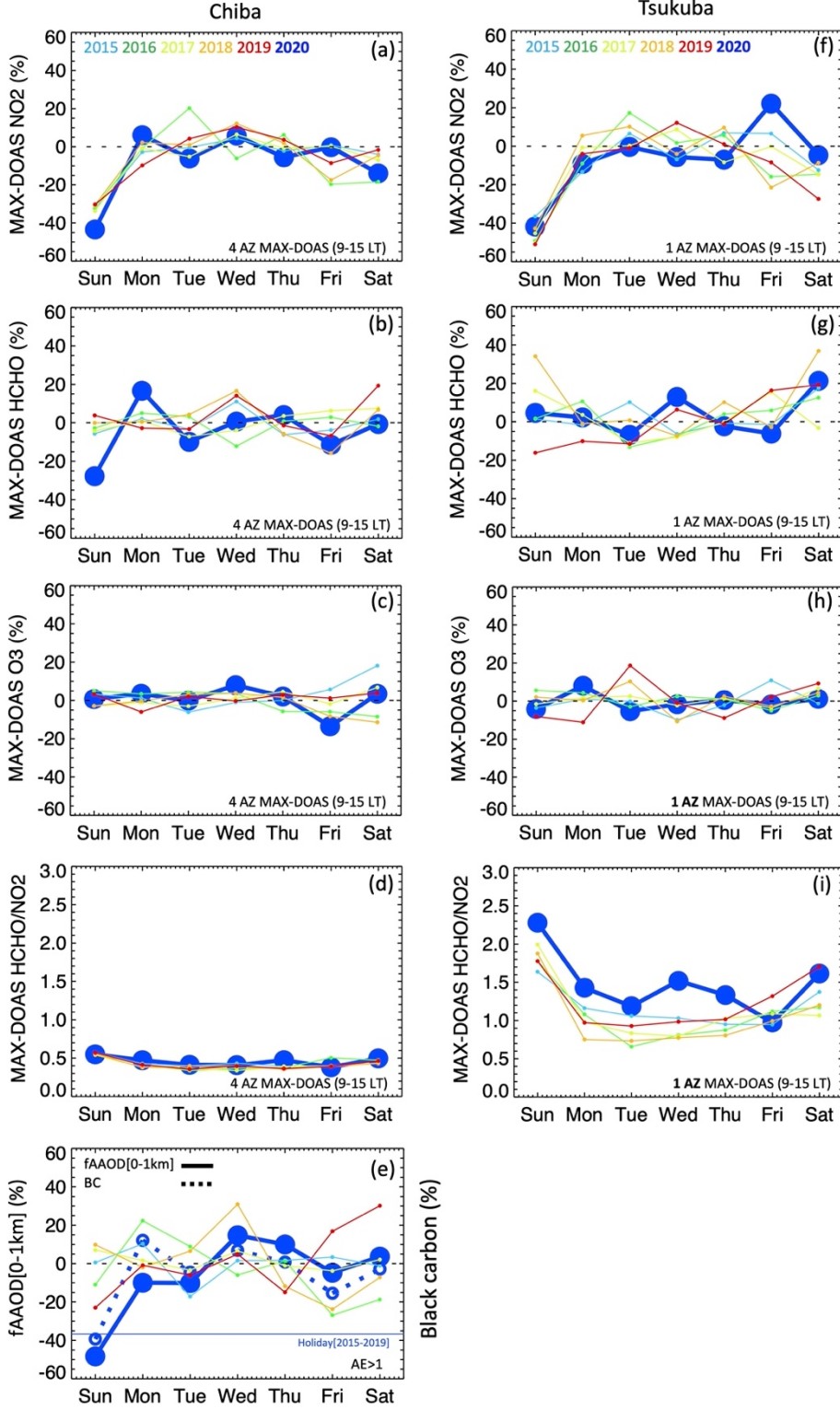








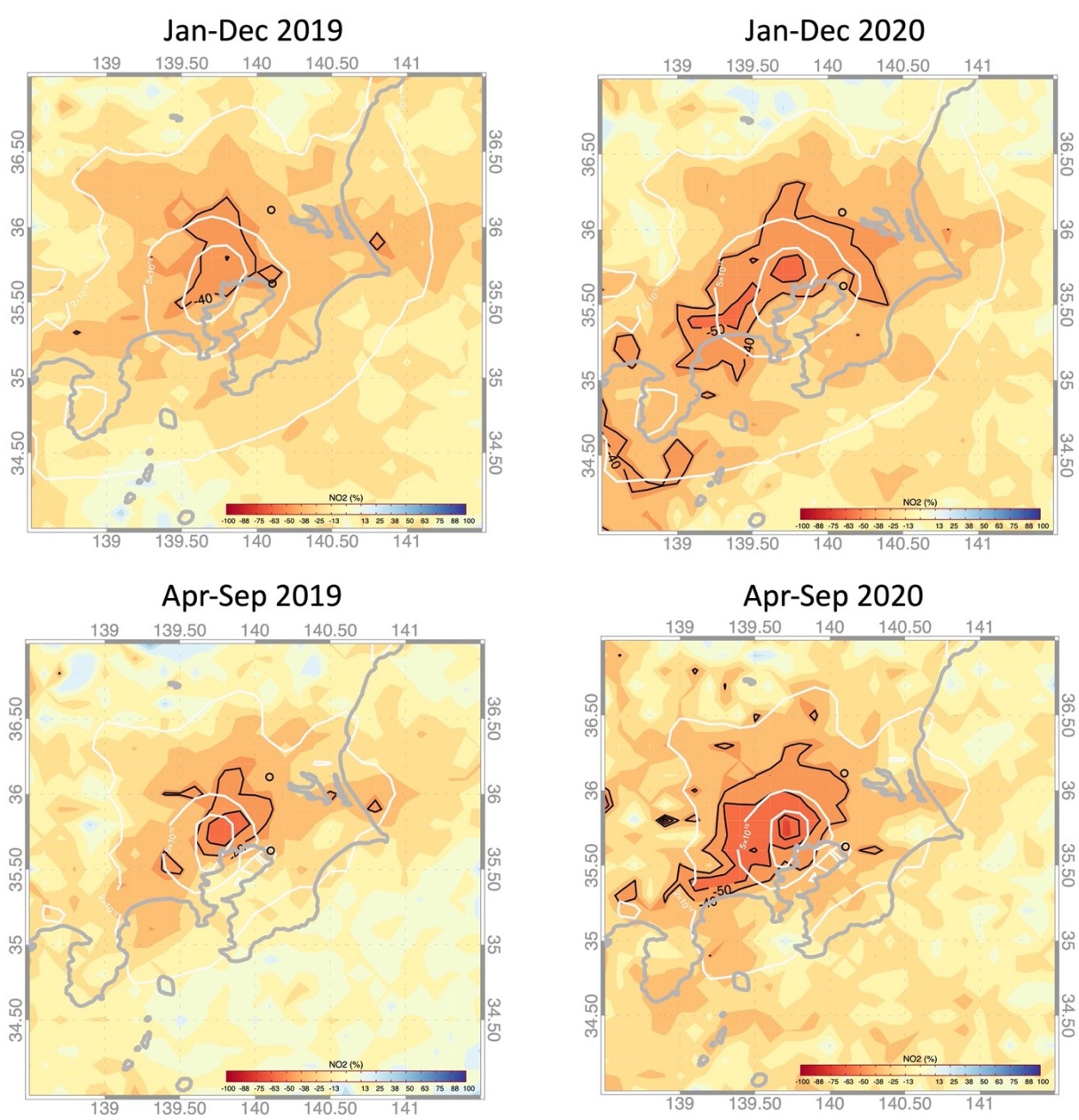

Figure 7: Weekly changes in MAX-DOAS partial tropospheric column (top to bottom) NO$_2$, HCHO, O$_3$, HCHO/NO$_2$, and fAAOD for Chiba (right) and Tsukuba (left) during 2015–2020. Only data with SZA < 50 were considered in the O$_3$ dataset; see text for details. Panel (e) includes the weekly cycle of surface BC concentrations. BC data were unavailable for Tsukuba.

**Figure 8: Spatial distribution of the TROPOMI NO₂ weekend effect (Sunday minus weekdays) in January to December 2019 (a), January to December 2020 (b), April to September 2019 (c), and April to September 2020 (d).**