# Peer review of "Peculiar COVID-19 effects in the Greater Tokyo Area revealed by spatiotemporal variabilities of tropospheric gases and light-absorbing aerosols"

_Atmospheric Chemistry and Physics, 2022_

## Author Comment (AC1)

**Reply to Reviews of "Peculiar COVID-19 effects in the Greater Tokyo Area revealed by spatiotemporal variabilities of tropospheric gases and light-absorbing aerosols" by Damiani et al.**

We thank reviewers for reading our paper and providing constructive comments. We have incorporated the reviewers' suggestions into the revised manuscript and provided point-by-point responses to each comment below. In the following, we first report the referees' comments (in black), then we provide our responses (in red). When included, line numbers refer to the revised manuscript.

**Anonymous Referee #1**

The authors focus on the spatiotemporal variability of gases and light-absorbing aerosols in the Greater Tokyo Area during the COVID19 lockdown and the resulting changes in mobility. In general, I find this manuscript to be of interest for publication and appropriate for Atmospheric Chemistry and Physics. The manuscript is well written and would benefit from some additional details in the methods and discussion sections. Consequently, I can only recommend this paper for publication after major revisions.

R -> We thank Reviewer 1 for taking the time to carefully reading our manuscript and providing many valuable comments. We addressed all points raised by the Reviewer, and we feel that now the manuscript is sensibly improved.

Major comments

The content of the manuscript needs to be backed by more references. I have mentioned some instances under minor comments.

R -> Thank you! We included all mentioned references (and others suggested by Reviewer 2) in the revised manuscript to support our findings.

The authors also need to provide an overview of other studies which have investigated the impact of COVID19 lockdown.

R -> We agree. Following your suggestion and the advice of Reviewer 2, we expanded the previous discussion on the impact of COVID-19 by focusing on the East-Asia region, which is more relevant for our study, as follows: (l64-73)

"After the first COVID-19 cases in Wuhan, to prevent the spread of the pandemic, strong social distancing and quarantine measures were implemented in many Chinese cities as early as 24 January 2020 till about 25 February; then, measures gradually downgraded to a partial lockdown. Evident decreases in most air pollutant concentrations have been reported for China, with satellite-based $NO_2$ reductions of about 40 % (Bauwens et al., 2020; Le et al., 2020; Levelt et al., 2022). In Wuhan, TROPOMI $NO_2$ and HCHO decreased to about 83 % and 11 % in February (Ghahremanloo et al., 2021). Comparable HCHO decreases were found in the Northern China Plain (Sun et al., 2021; Levelt et al., 2022). Moreover, surface observations showed a general increase in surface ozone in most of the regions, although ozone decreased in the subtropical south and, besides the reduced emissions, meteorological changes were found to be essential contributors (Sicard et al., 2020b; Le et al., 2020; Liu et al., 2021; Levelt et al., 2022). In Korea, the most significant changes occurred in March, with a reduction of about 20 % in $NO_2$ and 45 % in PM2.5 nationwide, while surface ozone, in contrast with China, was slightly decreased (Ju et al., 2021)."

The authors should also compare their findings with existing literature (Cooper et al., 2022, Miyazaki et al., 2021).

R -> Thank you! We included the additional discussions as follows: (l290-301)

"Based on satellite observations and model simulations, Cooper et al. (2022) estimated a significant overall decrease in surface $NO_2$ over more than 200 cities around the world in April 2020 compared with 2019. Among others, they reported $NO_2$ changes for various Japanese cities. Within the Kanto region, they showed reductions peaking at Yokohama (-69 %), more minor changes at Saitama (-32 %), and values roughly in between at Tokyo (-54 %). Despite the inverse correlation between the lockdown Stringency Index and the $NO_2$, they found that changes in Japan were comparable or slightly lower than those for the European cities where lockdown restrictions were much more stringent. In agreement with our findings (Fig. S1d and Fig. S3), they showed that changes in Japan could have been favored by meteorology and long $NO_2$ trends. Although the period examined by Cooper et al. (2022) only

partially coincides with the Japanese state of emergency, Fig. 3o shows comparable reductions. Moreover, Fig. 3o reveals the complex pattern of these variations, characterized by an evident North-South gradient with the most significant (negative) changes in Southern Tokyo, further evolving toward zero changes in the Saitama prefecture. This highlights the necessity of coupling detailed analysis at a regional scale with a large-scale study when examining COVID-related impacts, particularly when focusing on areas dominated by several megacities."
(l361-366)

"Miyazaki et al. (2021) showed that the pandemic caused a reduction in global NOx emissions resulting in an overall decreased free-tropospheric ozone and some isolated enhancements, due to the titration effect, at the surface in correspondence with strongly urbanized regions (mainly in China). Although they primarily focused on a global scale, so their study is hardly comparable with our findings at a regional scale, the expected change of ozone with the altitude in large urbanized areas suggests that MAX-DOAS ozone columns could result in negligible variations during the period of the emergency state resulting from summing such positive and negative changes within the column."

Line 128 There may have been a version change in TROPOMI products in this time period. If yes, the authors should briefly mention this and discuss how did they go about it.
R-> You are right. Various versions of the TROPOMI NO2 and HCHO products have been produced during the last years. Nevertheless, in this study, we used the official TROPOMI NO2 and HCHO products from the Tropospheric Emission Monitoring Internet Service and the Copernicus Open Access Hub, respectively. The TM5-MP-DOMINO NO2 dataset combines the versions 1.2.x. and 1.3.x. Version 1.3.x. was introduced on 2019/02/06, so, essentially, it covers the entire period here examined (i.e., January 2019–December 2020). Minor differences exist between the versions 1.2.x. and 1.3.x. and, according to all past studies, we combined them (Van Geffen et al., 2021). Similarly, the current TROPOMI HCHO product is based on version V2.1.3 after 2020/07/13 and version V1.1.x for the period before (De Smedt et al., 2021). We included this additional information in the revised manuscript.

Also, what do the authors mean by 'interpolation' here? Do they mean 'oversampling'? How have the uncertainties been considered?
R -> Please note that we did not oversample the data. Similar to previous studies (e.g., Barré et al., 2021; Ialongo et al., 2020), TROPOMI data were only binned and averaged over a regular grid to perform various statistical analyses at each location. We kept the grid box large enough (0.1 x 0.1 deg) to include a sufficient number of observations and examine the changes even on a daily base. To put it into context, consider that TROPOMI observations can be potentially oversampled to a grid of 0.01 x 0.01 deg with a one-month averaging period (Cooper et al., 2022 and references therein). We included this additional information in the revised manuscript.

Are the TROPOMI columns shown in Figure 3 and referred to in line 235 error-weighted averages?
Line 240-245 I would recommend the authors to also perform a sensitivity check if they considered median and error-weighted mean (if not already) of the TROPOMI HCHO columns and see if the interpretation of the results changes.
R-> We made more clear the original statement at line 235 as follows:
"Despite the high spatial heterogeneity of HCHO concentrations due to its short lifetime, the spatial distribution of the TROPOMI HCHO column was estimated (Panel (e–h) in Fig. 3).".
Panels in Figure 3 show mean values. When using an error-weighted mean, TROPOMI HCHO maps are essentially the same as those shown.

Can the authors use existing literature to comment on the relative contribution of biogenic and anthropogenic sources to HCHO and interpret the current findings of no apparent changes in TROPOMI HCHO?
R -> Yes, we included the following discussion:
(l304-307)
"The principal source of HCHO is the oxidation of methane, which provides a global ambient background (e.g., Surl et al., 2018). Then, over continental atmospheres, the main anthropogenic sources of HCHO are vehicle exhaust and industrial emissions, while the main natural sources are plants and biomass burning (Surl et al., 2018; Sun et al. 2021; Ghahremanloo et al., 2021)."
(l318-328)

"A summer maximum characterizes the observed seasonal cycle of the HCHO columns shown in Fig. 4g. This indicates that biogenic emissions dominate HCHO even within our urban region. Pieces of evidence in TROPOMI HCHO reductions as a consequence of the COVID-related mobility restrictions have been reported only for China (Ghahremanloo et al., 2021) while meteorology likely drove most of the HCHO variations in India (Levelt et al., 2022). However, even in Wuhan, while the reduction in NO2 reached about 83 %, the decrease in HCHO was only 11 %. The recent study by Sun et al. (2021) showed that comparable HCHO reductions (i.e., 11 %) were found in the Northern China Plain for locations with predominant declines in NO2 columns and elevated anthropogenic NMVOC emissions. However, reductions were favored by meteorological conditions. Then, simulations showed that most of the HCHO decrease resulted from the reduced anthropogenic NOx emissions. Still, an additional reduction in anthropogenic NMVOC emissions of about 15 % would be necessary to match the observations (Sun et al., 2021). Since mobility restrictions in Japan were less severe and more gradual than those established in China, we expect such minor HCHO variations hardly identifiable by using satellite observations."

Figure 5 It is difficult to read and interpret sub-figure (a) because of the image resolution. Also, the figure caption needs to describe that the NO2 mean is shown in grey. Can the authors consider an alternative way to present the data in this figure? For example, have the data as a table (supplementary material) and only show the top 10 or 20 cities in a figure (main text).
R -> Following your suggestion, to allow the reader to check the values better, we added the list of the cities with the associated NO2 changes shown in Fig 5a in the supplement (Tab. S1). Moreover, we corrected the caption by mentioning that the NO2 mean shown in grey. Nevertheless, in Fig. 5a, we prefer to avoid showing 10-20 cities only because this panel's primary goal is the visive comparison between the Japanese cities and the rest of the world. So, we did not modify panel (a), but we have provided a high-resolution figure for the revised manuscript, allowing for reading the details.

The contours in sub-figure (c) are also complex to interpret as the color legend is only for population density. Please describe in the caption how the contours should be interpreted.
R -> You are right. The caption was not informative enough.
The red contours show NO2 weekend-related changes, while the black contours show NO2 wind-related changes. Contours for both red and black solid lines read as follows: $-1,-2,-3,-4,-5,-6 \times 10^{15}$ molec./cm$^2$. Then, the black dashed lines show also positive changes (0.5, 1 $\times 10^{15}$ molec./cm2).
Now we included this additional information in the updated caption.

Minor comments

Lines 24-34 The introductory paragraph lacks references. More references for health effects of NO2 such as Achakulwisut et al. (2019), and for NO2 trends using satellite data such as Vohra et al. (2022).
R-> We included both of them.

Lines 35-40 Have the emissions of ozone precursors significantly decreased worldwide? I do not suspect the same in Asia and Africa. The authors should mention whether this refers to any city or a larger region and if both VOCs and NOx emissions are decreasing? It would be a good idea for authors to add details of the studies cited.
R -> Thank you for raising this point. We have elaborated better this statement as follows:
"Indeed, in recent years, satellite observations showed that, although NOx emissions are still rising in various developing countries (e.g., India), they have significantly decreased in the majority of the developed countries of North America, Europe, and East Asia (Russell et al., 2012; Geddes et al., 2016; Georgoulias et al., 2019), while tropospheric ozone has increased (Ziemke et al., 2019; Li et al., 2019; Lee et al., 2021)."

Line 91 Reference for HCHO as a proxy for VOCs.
R-> we included the pioneering study of Sillman (1995)

Line 112 Which year are the MAX-DOAS measurements from? Refer to line 185.
R -> The periods are 2013-2020 for Chiba and 2015-2020 for Tsukuba, but there are no MAX-DOAS ozone observations in winter (see Sect. 2.1.1), so, in Fig 5e (and, for consistency, in Fig. S2d) we used ozonesonde data

recorded during the same years. We made this more understandable in the revised manuscript by adding some details in both Section 2.1.1 and Section 2.1.7.

Line 113 Any reference to support this?
R -> yes, we included Takashima et al. (2009)

Line 133 Any reference to support this? Why is the cloud fraction criteria different from OMI? Is it possible to assess the impact on the results if this cloud fraction threshold were to be changed to 0.3?
R -> As detailed in the text, OMI NO2 and TROPOMI NO2 datasets are based on slightly different retrieval algorithms, including cloud algorithms to estimate cloud fraction. Moreover, the size of the field of view of OMI is larger than that of TROPOMI, and OMI is affected by the row anomaly problem (http://omi.fmi.fi/anomaly.html), which further reduces the number of available observations. Here, the slightly larger threshold for OMI (i.e., 0.3 vs. 0.2 for OMI and TROPOMI, respectively) potentially tends to counteract the smaller number of available OMI observations. Generally, most of the past studies based on OMI and TROPOMI used cloud fractions within the range of 0.3-0.5 (Goldberg et al., 2021; Wang et al., 2020). Similar thresholds were employed for satellite validation studies with ground observations, such as the recent global validation of TROPOMI (Verhoelst et al., 2021), which included our ground-based MAX-DOAS observations here used. However, it is expected that the lower the cloud fraction is, the more accurate NO2 observations are, considering that too low a cloud fraction can reduce the data records excessively, especially in some regions and seasons.
A study that evaluates the impact of the cloud fraction on the quality of OMI and TROPOMI NO2 columns could be carried out by comparing satellite observations with ground-based MAX-DOAS observations. However, such a potential analysis, which is beyond the objectives of the present study, would be complicated by the influence of additional factors like aerosol load and surface albedo and could hardly highlight appreciable differences within the small threshold range of 0.2–0.3.
We included some additional details as above in the revised manuscript.

Line 177 OMI overpass time earlier is 13:40 LT and here it is 1:30pm. The authors should use consistent values and formats.
R -> Corrected.

Line 212-216 Confusing. Please rephrase.
R -> We rephrased and expanded this paragraph as follow:
(l258-270)
"Under days characterized by stagnant low wind speed conditions, NO2 accumulates around source locations. In contrast, under days with high wind speed conditions, NO2 is dispersed. Tokyo is located in a polluted background with various significant NOx sources surrounding it within about a 100 km radius. Therefore, due to the influence of surrounding sources, the outflow plume of NO2 from Tokyo is not evident in the TROPOMI NO2 maps. The spatial pattern of the difference between these two NO2 composites, built based on wind speed data, reveals outflow patterns more clearly (see also Liu et al., 2016). We applied this method limitedly to Fig. 5c. To select the threshold values to identify high and low wind speed days for each pixel, we used MERRA-2 wind fields. According to previous studies (e.g., Fioletov et al., 2022), we used a PBL averaged wind. Still, the results are not sensitive to the wind altitude because the wind is relatively constant within the boundary layer. Composite differences between high and low wind speed days in TROPOMI NO2 were computed based on MERRA-2 wind fields averaged around the overpass time (12–3 pm). The median wind speed of each pixel was assumed to be the threshold between the high and low wind composite values. We first regridded the MERRA-2 data to the resolution of TROPOMI; then, for each grid cell, we computed NO2 as the difference between the composite values of days with high and low wind speed."

Line 240 Is there an increasing trend in CH4 which could be playing a role here?
R-> Due to the long atmospheric lifetime, CH4 is not expected to be impacted by lockdown measures. Long-lived VOCs like methane contribute to the background levels of HCHO, whereas the spatial variability of HCHO is related to shorter-lived NMVOCs. In the continental boundary layer, other natural short-lived VOCs (with isoprene being often the dominant precursor) increase HCHO concentrations over the background levels. Indeed, the observed seasonal cycle of the HCHO column shown in Fig. 4g points to the fact that biogenic emissions dominate

HCHO. Since methane oxidation is the dominant source of HCHO in the remote atmosphere, if the increasing trend in CH4 would play some role, we should also be able to identify patterns in the HCHO difference between 2020 and 2019 over the ocean. Instead, HCHO does not show any pattern over the sea. (Fig. 3n).

Line 249-250 There needs to be some discussion around what the value of this ratio is. What is the transition regime you have been considering given this varies with both space and time (Duncan et al., 2010)? Add references too.
R -> OK! In the revised manuscript we added the following discussion:
(l329-338)
"Traditionally, the ozone production regime is considered to be VOC-limited when this ratio is lower than 1, NOx-limited when it is higher than 2, while ozone is expected to be in the transition regime when the values are in the range 1–2 (Duncan et al., 2010; Ryan et al., 2020). Although several studies used this ratio to infer O3 sensitivity to NOx and VOCs by using observations from satellite and ground-based instruments (Duncan et al., 2010; Jin et al., 2017; Schroeder et al., 2017; Irie et al., 2021), some limitations still exist. Assuming the transition region lies within the range 1–2 (Duncan et al., 2010) could not be valid at global levels, and it could be necessary to compute it depending on the region (Schroeder et al., 2017). Moreover, the ratio has an altitude dependence (e.g., Jin et al., 2017; Schroeder et al., 2017). While seasonal variations and trends in the columnar HCHO/NO2 ratio (i.e., based on satellite observations) generally match the ratio computed with in situ observations, magnitudes are often different due to different vertical distributions of HCHO and NO2 (Ryan et al., 2020)."

Line 275 Were the CAMS measurements for 2020 not available at the time of analysis? If they are available now, they should be included in the study.
R -> CAMS data for 2020 were available and we showed all data in Fig. 4.
By mentioning that "anomalous emissions that occurred in 2020 are not included" we mean that CAMS simulations, used in this work, are based on business-as-usual emissions scenario, therefore did not account for the expected emission reduction due to COVID. We included additional details in the revised manuscript to avoid misunderstandings.

Line 289  Earlier in line 145, the OMI data record is mentioned as 2005-2019.
-> Thank you for pointing out that! We used OMI data in 2005-2020. We corrected that at line 145.

Line 290-291 How many cities have been removed because of Friday being a rest day? The authors should list them either in the main text or in the supplementary for completeness. Also, if a lot of cities have been removed, the authors can consider the weekdays to be from Monday to Thursday and compare with Sunday for the weekend effect.
R -> The majority of the large cities with Friday as a rest day (all located in the Middle East) show medium/small reductions, and they would not enter the ranking. Only Mecca and a few others in Saudi Arabia could be included but toward the bottom, so they have a scarce interest within the framework of our study.
In the revised manuscript, now we remind the redear the recent work of Stavrakou et al. (2020) showing weekend NO2 changes on both Sunday and Friday worldwide, including a detailed list of these cities.
Moreover, we included the cities and NO2 changes shown in Fig. 5a in the new Tab. S1 of the supplementary.

Line 305 What are the "two contour lines"?
R -> the absolute difference in NO2 resulting from Sunday minus weekdays (red) and the absolute difference in NO2 resulting from high wind speed minus low wind speed days (black).
In the revised manuscript we included additional details to make it more understandable.

Line 412 How does this look compared to findings from Miyazaki et al. (2021)?
We included the following discussion in the revised manuscript:
(l361-366)
"Miyazaki et al. (2021) showed that the pandemic caused a reduction in global NOx emissions resulting in an overall decreased free-tropospheric ozone and some isolated enhancements, due to the titration effect, at the surface in correspondence with strongly urbanized regions (mainly in China). Although they primarily focused on a global scale, so their study is hardly comparable with our findings at a regional scale, the expected change of ozone with the altitude in large urbanized areas suggests that MAX-DOAS ozone columns could result in negligible variations

during the period of the emergency state resulting from summing such positive and negative changes within the column."

Figure 3 The color scale for the last column should be reversed. Warm colors should indicate positive values and cool colors negative.
R -> We prefer keeping the current color scale for consistency with the maps shown in figures 5 and 8. The use of cool colors for negative changes prevents highlighting the several overlapping isolines we included.

Figure 4 Caption text "Results are shown as percentage changes with respect to the 2013–2019 average (left and central panels) and 2019 (right panel)." is confusing. Please rephrase.
R -> We changed the caption as follows:
"Results are shown as percentage changes with respect to the 2013–2019 average (left and central panels) and with respect to 2019 (right panel)."

Figure 6 OMI total columns are referred to in the figure but tropospheric columns in the caption. Please correct as needed.
R-> In the figure, OMI NO2 total column means Total tropospheric column to differentiate it from the NO2 partial column provided by MAX-DOAS. We corrected the caption.

For data products (OMI/TROPOMI/MAX-DOAS, etc), please include URL stating when they were last accessed or point to references if data not publicly available, so that potential users can use these.
R-> We updated the URL for all datasets in the "Data availability" section. All datasets are publicly available.

References
Achakulwisut et al., doi: 10.1016/S2542-5196(19)30046-4, 2019.
Cooper et al., doi:10.1038/s41586-021-04229-0, 2022.
Duncan et al., doi: 10.1016/j.atmosenv.2010.03.010, 2010.
Miyazaki et al., doi: 10.1126/sciadv.abf7460, 2021.
Vohra et al., doi:10.1126/sciadv.abm4435, 2022.
R -> Thank you. We included and discussed all references.

Additional references
- Barré, J., Petetin, H., Colette, A., Guevara, M., Peuch, V.-H., Rouil, L., Engelen, R., Inness, A., Flemming, J., Pérez García-Pando, C., Bowdalo, D., Meleux, F., Geels, C., Christensen, J. H., Gauss, M., Benedictow, A., Tsyro, S., Friese, E., Struzewska, J., Kaminski, J. W., Douros, J., Timmermans, R., Robertson, L., Adani, M., Jorba, O., Joly, M., and Kouznetsov, R.: Estimating lockdown-induced European NO2 changes using satellite and surface observations and air quality models, Atmos. Chem. Phys., 21, 7373–7394, https://doi.org/10.5194/acp-21-7373-2021, 2021.
- Brancher M., Increased ozone pollution alongside reduced nitrogen dioxide concentrations during Vienna's first COVID-19 lockdown: Significance for air quality management, Environmental Pollution, Volume 284, 2021, 117153, https://doi.org/10.1016/j.envpol.2021.117153.
- Duncan, B. N., Yoshida, Y., Olson, J. R., Sillman, S., Martin, R. V., Lamsal, L., Hu, Y. T., Pickering, K. E., Retscher, C., Allen, D. J., and Crawford, J. H.: Application of OMI observations to a space-based indicator of NOx and VOC controls on surface ozone formation, Atmos. Environ., 44, 2213–2223, https://doi.org/10.1016/j.atmosenv.2010.03.010, 2010.
- Fioletov, V., McLinden, C. A., Griffin, D., Krotkov, N., Liu, F., and Eskes, H.: Quantifying urban, industrial, and background changes in NO2 during the COVID-19 lockdown period based on TROPOMI satellite observations, Atmos. Chem. Phys., 22, 4201–4236, https://doi.org/10.5194/acp-22-4201-2022, 2022.
- Guevara M., O. Jorba, A. Soret, H. Petetin, D. Bowdalo, K. Serradell, C. Tena, H. Denier van der Gon, J. Kuenen, V.-H. Peuch, C. Pérez García-Pando, Time-resolved emission reductions for atmospheric chemistry modelling in Europe during the COVID-19 lockdowns, Atmos. Chem. Phys., 21 (2021), pp. 773-797, 10.5194/acp-21-773-2021

- Itahashi, S., Yamamura, Y., Wang, Z. et al. Returning long-range PM2.5 transport into the leeward of East Asia in 2021 after Chinese economic recovery from the COVID-19 pandemic. Sci Rep 12, 5539 (2022). https://doi.org/10.1038/s41598-022-09388-2
  Jin, X., Fiore, A. M., Murray, L. T., Valin, L. C., Lamsal, L. N., Duncan, B., Boersma, K.F., De Smedt, I., Abad, G.G., Chance, K., and Tonnesen, G. : Evaluating a space-based indicator of surface ozone-NOx-VOC sensitivity over midlatitude source regions and application to decadal trends. J. Geophys. Res., 122(19), 10,439-410,461, https://doi.org/10.1002/2017JD026720, 2017
- Ju M. J., J. Oh, Y. H. Choi et al., Changes in air pollution levels after COVID-19 outbreak in Korea, Science of the Total Environment 750, 141521, https://doi.org/10.1016/j.scitotenv.2020.141521, 2021
- Laughner J. L. et al., Societal shifts due to COVID-19 reveal large-scale complexities and feedbacks between atmospheric chemistry and climate change, 2021, 118 (46) e2109481118, https://doi.org/10.1073/pnas.2109481118
- Le T, Wang Y, Liu L, Yang J, Yung YL, Li G, Seinfeld JH. Unexpected air pollution with marked emission reductions during the COVID-19 outbreak in China. Science. 2020, 369(6504):702-706. doi: 10.1126/science.abb7431
- Levelt, P. F., Stein Zweers, D. C., Aben, I., Bauwens, M., Borsdorff, T., De Smedt, I., Eskes, H. J., Lerot, C., Loyola, D. G., Romahn, F., Stavrakou, T., Theys, N., Van Roozendael, M., Veefkind, J. P., and Verhoelst, T.: Air quality impacts of COVID-19 lockdown measures detected from space using high spatial resolution observations of multiple trace gases from Sentinel-5P/TROPOMI, Atmos. Chem. Phys., https://doi.org/10.5194/acp-2021-534, 2022 (in press)
- Liu et al., Diverse response of surface ozone to COVID-19 lockdown in China, Science of the Total Environment, 789, 147739, 2021, https://doi.org/10.1016/j.scitotenv.2021.147739
- Liu, F., Beirle, S., Zhang, Q., Dörner, S., He, K., and Wagner, T.: NOx lifetimes and emissions of cities and power plants in polluted background estimated by satellite observations, Atmos. Chem. Phys., 16, 5283–5298, https://doi.org/10.5194/acp-16-5283-2016, 2016.
- Shakil, MH, Munim, ZH, Tasnia, M, Sarowar, S. 2020. COVID-19 and the environment: A critical review and research agenda. Science of the Total Environment 745, DOI: http://dx.doi.org/10.1016/j.scitotenv.2020.141022.
- Shen, L., Jacob, D. J., Liu, X., Huang, G., Li, K., Liao, H., and Wang, T.: An evaluation of the ability of the Ozone Monitoring Instrument (OMI) to observe boundary layer ozone pollution across China: application to 2005–2017 ozone trends, Atmos. Chem. Phys., 19, 6551–6560, https://doi.org/10.5194/acp-19-6551-2019, 2019.
- Sillman, S.: The use of NOy , H2 O2 , and HNO3 as indicators for Ozone-NOx -Hydrocarbon sensitivity in urban Locations, J. Geophys. Res. Atmos., 100, 14175–14188, https://doi.org/10.1029/94jd02953, 1995
- Surl, L., Palmer, P. I., and González Abad, G.: Which processes drive observed variations of HCHO columns over India?, Atmos. Chem. Phys., 18, 4549–4566, https://doi.org/10.5194/acp-18-4549-2018, 2018
- Takashima H, Irie H, Kanaya Y, Shimizu A, Aoki K, Akimoto H (2009) Atmospheric aerosol variations at Okinawa Island in Japan observed by MAX-DOAS using a new cloud screening method. J Geophys Res 114(D18):D18213. https://doi.org/10.1029/2009JD011939
- Verhoelst, T., et al.: Ground-based validation of the Copernicus Sentinel-5P TROPOMI NO2 measurements with the NDACC ZSL-DOAS, MAX-DOAS and Pandonia global networks, Atmos. Meas. Tech., 14, 481–510, https://doi.org/10.5194/amt-14-481-2021, 2021.

**Anonymous Referee #2**

The paper by Damiani et al. is well structured and well written, with English of high quality. The paper has high-quality and informative figures. Combining different type of measurements for multiple species with model outputs and weather information provides a very complete record of changes in composition during lock-down, weekends and end-of-year holidays. I am in favour of publishing this paper after my major and minor comments have been addressed by the authors.
R -> We thank Reviewer 2 for the overall positive report and the useful suggestions. Below, we addressed all points raised.

Major comment:
In general I am of the opinion that the list of references does not well reflect the detailed studies conducted to document the COVID-19 lockdown impact on air pollution levels in the past two years. The authors could add reviews on this topic, like Gkatzelis et al., https://doi.org/10.1525/elementa.2021.00176
R-> We agree with the suggestion of the reviewer. So, we added additional reviews as follows:

- Gkatzelis G. I. et al., The global impacts of COVID-19 lockdowns on urban air pollution: A critical review and recommendations, Elementa: Science of the Anthropocene, 9 (1): 00176, 2021
- Shakil, MH, et al., COVID-19 and the environment: A critical review and research agenda. Science of the Total Environment 745. DOI: http://dx.doi.org/10.1016/j. Scitotenv.2020.141022, 2020.

add some extra citations about the interaction between ozone, NOx and aerosol during the lockdowns.
R-> Following your suggestion and the advices of Reviewer #1, we included extra citations as follows:

- Laughner J. L. et al.: Societal shifts due to COVID-19 reveal large-scale complexities and feedbacks between atmospheric chemistry and climate change, 118 (46) e2109481118, 2021
- Cooper, M. J., et al.: Global fine-scale changes in ambient NO2 during COVID-19 lockdowns, Nature, 601, 380-387, doi:10.1038/s41586-021-04229-0, 2022.
- Miyazaki, K. et al.: Global tropospheric ozone responses to reduced NOx emissions linked to the COVID-19 worldwide lockdowns, Science Advances, 7, 24, doi: 10.1126/sciadv.abf7460, 2021.
- Several additional references related to lockdowns in East Asia (see below) and others (see Additional references below)

The authors remark that "many studies" on the relation COVID-19 and air quality have been conducted in the past two years, including results for the country of Japan. The authors should cite more extensively papers discussing the East-Asia region to provide the reader with a good overview on what is already published on COVID-19.
R-> Thank you for your suggestion! We added a further discussion about the COVID-related changes occurred in East Asia as follows:
(l64-73)
"After the first COVID-19 cases in Wuhan, to prevent the spread of the pandemic, strong social distancing and quarantine measures were implemented in many Chinese cities as early as 24 January 2020 till about 25 February; then, measures gradually downgraded to a partial lockdown. Evident decreases in most air pollutant concentrations have been reported for China, with satellite-based NO2 reductions of about 40 % (Bauwens et al., 2020; Le et al., 2020; Levelt et al., 2022). In Wuhan, TROPOMI NO2 and HCHO decreased to about 83 % and 11 % in February (Ghahremanloo et al., 2021). Comparable HCHO decreases were found in the Northern China Plain (Sun et al., 2021; Levelt et al., 2022). Moreover, surface observations showed a general increase in surface ozone in most of the regions, although ozone decreased in the subtropical south and, besides the reduced emissions, meteorological changes were found to be essential contributors (Sicard et al., 2020b; Le et al., 2020; Liu et al., 2021; Levelt et al., 2022). In Korea, the most significant changes occurred in March, with a reduction of about 20 % in NO2 and 45 % in PM2.5 nationwide, while surface ozone, in contrast with China, was slightly decreased (Ju et al., 2021)."

Starting from this the authors should subsequently indicate what is new in the present work, and how this complements the earlier studies.

R -> We pointed out how our work complement others and what is new as follows:
(l98-115)
"As detailed above, many previous studies examined COVID-related changes in air quality on a global to local scale. Nevertheless, due to somewhat soft countermeasures to limit the spread of the pandemic adopted in Japan with consequent more limited change in the mobility compared with other countries, relatively fewer studies focused on this area (e.g., Itahashi et al., 2022). In some cases, changes in relevant air-quality parameters observed by ground-based or satellite instruments in the Tokyo center during the emergency period have been examined on a local scale (Sugawara et al., 2021) or related to other cities/countries on a global scale (e.g., Cooper et al., 2022) within studies aimed at comparing such variabilities with mobility changes. Nevertheless, as we will see, such changes hide a sizeable spatiotemporal variability and a widespread adherence to recommendations designed to limit the spread of the pandemic, which caused modification of common habits. Those resulted in a unique air quality signature not limited only to the emergency period, which should be examined on a regional scale.
We focused our study on the Greater Tokyo Area (GTA) in the Kanto region, which is the largest flat land in Japan, extending inland from the Pacific coast (Fig. 2). It is the most populous metropolitan area in the world and the most important economic hub of East Asia, and local emissions dominate it. Most of this large urban area is expected to be under VOC-limited conditions (Akimoto, 2017; Irie et al., 2021). Nevertheless, western Japan and, to a lesser extent, this region are usually affected by transboundary air pollution from the continent (Itahashi et al., 2022). Due to the strict mobility restrictions implemented in China, this additional contribution is expected to be reduced in early 2020 (Itahashi et al., 2022). This makes the analysis of the COVID-related effects even more complex and points to the necessity of a regional study focusing on spatiotemporal variability."

Minor remarks:
Abstract :
l15: "NO2 concentrations". It would be good to mention if this refers to surface, lower troposphere, column or all. Same for aerosol.
-> We included this information in the abstract of the revised manuscript

l18: Maybe better remove "in recent years", or do the authors mean that this happens both in 2021 and 2020?
-> Removed

Figure 1: The time axis (x-axis labels) in panel (a) is difficult to interpret: 2020.4 seems to coincide with the end of May. Would be useful to have 12 major ticks with months "Jan", "Feb" etc. For panel (b) could you please indicate that the period 7 April - 25 May was used. How is the 0% level determined?
R -> Following your suggestion, we included the new labels in panel (a) and additional information in the caption as follows: The baseline in both panel (a) and panel (b) is the median value for the corresponding day of the week during the 5-week period of January 3–February 6, 2020. Also, Panel (b) is based on the period Feb 7–Dec 31, 2020.

l90: "In this study, we apply an integrated approach ..". See my general comment: why is this study unique, and what new result(s) are obtained?
R -> We agree. As mentioned above, we included the following text:
(l98-115)
"As detailed above, many previous studies examined COVID-related changes in air quality on a global to local scale. Nevertheless, due to somewhat soft countermeasures to limit the spread of the pandemic adopted in Japan with consequent more limited change in the mobility compared with other countries, relatively fewer studies focused on this area (e.g., Itahashi et al., 2022). In some cases, changes in relevant air-quality parameters observed by ground-based or satellite instruments in the Tokyo center during the emergency period have been examined on a local scale (Sugawara et al., 2021) or related to other cities/countries on a global scale (e.g., Cooper et al., 2022) within studies aimed at comparing such variabilities with mobility changes. Nevertheless, as we will see, such changes hide a sizeable spatiotemporal variability and a widespread adherence to recommendations designed to limit the spread of the pandemic, which caused modification of common habits. Those resulted in a unique air quality signature not limited only to the emergency period, which should be examined on a regional scale.
We focused our study on the Greater Tokyo Area (GTA) in the Kanto region, which is the largest flat land in Japan, extending inland from the Pacific coast (Fig. 2). It is the most populous metropolitan area in the world and the most important economic hub of East Asia, and local emissions dominate it. Most of this large urban area is expected to be under VOC-limited conditions (Akimoto, 2017; Irie et al., 2021). Nevertheless, western Japan and, to a lesser extent, this region are usually affected by transboundary air pollution from the continent (Itahashi et al., 2022). Due

to the strict mobility restrictions implemented in China, this additional contribution is expected to be reduced in early 2020 (Itahashi et al., 2022). This makes the analysis of the COVID-related effects even more complex and points to the necessity of a regional study focusing on spatiotemporal variability."

l105: "FWHM = 0.4 nm at 357 and 476 nm". Why mention these two wavelengths instead of saying something like ""FWHM = 0.4 nm for this wavelength range". Is there a large change in FWHM as a function of wavelength?
R -> No, we only mentioned them because they are the nominal wavelengths of the aerosol retrieval algorithm. We rephrased the sentence as follows:
"High-resolution spectra were recorded from 310 to 515 nm using the Maya2000Pro spectrometer (Ocean Insight, Inc., Orlando, FL, USA) with a slit of 25 μm and a full width at half maximum of approximately 0.3–0.4 nm, enclosed in a temperature-controlled box."

l106: "wavelength calibration was performed daily to account for .. signal degradation" ? Do you mean "radiometric calibration" ?
R -> We rephrased the sentence as follows:
"Wavelength calibration was performed daily, using a high-resolution solar spectrum, to account for potential long-term degradation of the spectrometer."

l113: "relative humidity over water ". Why "over water"?
R -> Usually, for temperatures over 0°C relative humidity is calculated for saturation over water (in contrast to saturation over ice). Diurnal temperatures in the Kanto region are consistently over 0°C (also in winter).

l115: "This procedure is expected to better account". Can this be tested, e.g. by comparing the four measurements?
R -> Yes. For example, inset panels of the supplementary Fig. S3 show the spatial heterogeneity of NO2 at our station as measured by our ground-based system. Overall, there is a North-South gradient which is also captured by TROPOMI observations. This was discussed around l383 of the original manuscript.

ll119: "but sampled at higher accuracy". Please explain.
R-> Generally, well calibrated and traceable ground-based observations based on previously well validated retrieval algorithms constitute the ground true against to satellite observations are evaluated. Because of their higher accuracy, and their spatial resolution comparable to the satellite pixel, MAX-DOAS observations are particularly suitable for such validation exercises.
Within this framework, it is worth to highlight that our MAX-DOAS observations contributed to recent efforts of the scientific community to validate TROPOMI NO2 and HCHO datasets at a global scale (Verhoelst et al., 2021; De Smedt et al., 2021).
We included an additional statement as above in the revised manuscript.

l127: "we used the NO2 and HCHO datasets". Please provide the processor versions of both datasets.
R -> we included the respective versions and further minor details in the revised manuscript.

l128: "interpolated over a regular grid of 0.1 × 0.1°". Why was this done? One extra interpolation step will potentially degrade the comparison, adding extra representativity uncertainty.
R -> analogy to previous works (e.g., Barré et al., 2021; Ialongo et al., 2020), the TROPOMI data were binned and averaged on a regular grid to perform statistical analyses and evaluate the data at various time scales at each location (i.e., grid box).

l132: "Screening of TROPOMI NO2 data involved retaining data with a quality flag (QF) value higher than 0.5 and a cloud fraction (CF) lower than 0.2." The README file of TROPOMI suggests the removal of data with a quality value < 0.75. Why did the authors use a different filtering?
The TROPOMI ATBD recommends a quality flag of either 0.75 or 0.5, depending on the specific applications. A quality flag of 0.75 automatically removes clouds with cloud fraction > 0.5, snow-covered scenes, and other problematic retrievals. However, a quality flag of 0.5 is also proposed as good enough. It includes good quality retrievals over clouds and snow/ice scenes and has been frequently adopted (e.g., Eskes and Eichmann, 2019; Huang and Sun, 2020; Kawka et al., 2021), allowing a more significant number of retained observations to be analyzed. This is often necessary, for example, to calculate reliable monthly averages for winter or, as in our case, monsoon

seasons. Then, being snow essentially absent even in winter and aerosol load relatively low, clouds are the major issue at our location; so, it should be noted that we further imposed a cloud fraction threshold slightly more stringent than the usual one implicitly retaining observations with higher quality flag.

Why is the cloud fraction limit different from OMI?
R -> As detailed in the text, OMI NO2 and TROPOMI NO2 datasets are based on slightly different retrieval algorithms, including cloud algorithms to estimate cloud fraction. Moreover, the size of the field of view of OMI is larger than that of TROPOMI, and OMI is affected by the row anomaly problem (http://omi.fmi.fi/anomaly.html), which further reduces the number of available observations. Here, the slightly larger threshold for OMI (i.e., 0.3 vs. 0.2 for OMI and TROPOMI, respectively) potentially tends to counteract the smaller number of available OMI observations. Generally, most of the past studies based on OMI and/or TROPOMI used cloud fractions within the range of 0.3-0.5 (e.g., Goldberg et al., 2021; Wang et al., 2020). Similar thresholds are employed for satellite validation studies with ground observations such as the recent global validation of TROPOMI (Verhoelst et al., 2021) which included our ground-based MAX-DOAS observations here used. However, it is expected that the lower the cloud fraction is, the more accurate NO2 observations are, taking into account that too low a cloud fraction can reduce the data records excessively, especially in some regions and seasons.

l187: I assume that wind, PBL height and temperature are also available in the CAMS reanalysis data record? Why do the authors use also MERRA? Does this have advantages over CAMS?
R-> Here, we used MERRA data because these MERRA-based parameters are usually more frequently used than the correspondent CAMS products. This is also true for most COVID-related papers available in the literature. Therefore, using MERRA allows a more straightforward comparison with previous results.

l196: Could you please explain what "transit stations" means. Is this bus and train only? Road traffic would be more relevant for emissions I guess. Does the transit station class scale well with the number of cars and trucks?
R-> Yes, it corresponds to mobility trends for places like public transport hubs such as subway, bus, and train stations. Following previous studies (e.g., Guevara et al., 2021), we used Google mobility data as a proxy for traffic counts as they are easily accessible for the majority of the countries and allowed us to compare the changes that occurred in different regions (Fig. 1).
Google transit data has been previously used to estimate the emission reduction for the road transport sector (Guevara et al., 2021). It assumes that mobility trends in public transport hubs can be taken as a proxy for trends in road traffic emissions. This assumption is likely more appropriate for lighter vehicles than for heavier vehicles (Brancher, 2021).
We included this additional information in Methods.

l206: "were estimated to be described" Please reformulate.
R-> We changed the sentence as follows:
"Light-absorbing aerosols within the boundary layer were estimated by combining sky radiometer and MAX-DOAS optical property data at UV wavelengths (Damiani et al., 2021)."

l215: I found this part very difficult to understand. At which altitude was the wind speed sampled? Is it the 10m wind, or PBL averaged wind, or something else? What is high, and what is low wind? "we computed NO2 as the difference between the composite values of days with high and low wind speed." Please explain the logic behind this. What does this difference represent? Is the difference plotted in Fig. 3, or the TROPOMI NO2 value itself?
R-> Please note that we applied this method exclusively to Fig. 5c. Fig. 3 shows the distribution of the gases in different years and their difference (2020 minus 2019). In the revised manuscript we explicitly mentioned that and made the method clear as follows:
(l258-270)
"Under days characterized by stagnant low wind speed conditions, NO2 accumulates around source locations. In contrast, under days with high wind speed conditions, NO2 is dispersed. Tokyo is located in a polluted background with various significant NOx sources surrounding it within about a 100 km radius. Therefore, due to the influence of surrounding sources, the outflow plume of NO2 from Tokyo is not evident in the TROPOMI NO2 maps. The spatial pattern of the difference between these two NO2 composites, built based on wind speed data, reveals outflow patterns more clearly (see also Liu et al., 2016). We applied this method limitedly to Fig. 5c. To select the threshold values to identify high and low wind speed days for each pixel, we used MERRA-2 wind fields. According to

previous studies (e.g., Fioletov et al., 2022), we used a PBL averaged wind. Still, the results are not sensitive to the wind altitude because the wind is relatively constant within the boundary layer. Composite differences between high and low wind speed days in TROPOMI NO2 were computed based on MERRA-2 wind fields averaged around the overpass time (12–3 pm). The median wind speed of each pixel was assumed to be the threshold between the high and low wind composite values. We first regridded the MERRA-2 data to the resolution of TROPOMI; then, for each grid cell, we computed NO2 as the difference between the composite values of days with high and low wind speed."

l221. What are "communication routes"? Do you mean "transportation routes"?
R-> yes, thank you!

l240: "does not align the urbanized region" -> does not align with the urbanized region
R-> thank you

l242: "application of cloud screening ". The filtering of the data follows the TROPOMI readme file. Does this remark mean that an additional or reduced cloud screening was applied on top of the standard filtering? Please explain what was done. "somewhat different": what is the reference here?
R-> No, the filtering of the data follows the TROPOMI readme file.
Here, we essentially refer to the effect of meteorology on the observations. In the revised manuscript we rephrased this sentence as follows:
"However, the change in meteorological conditions and the application of cloud screening cause the amount of data collected under clear sky conditions to be slightly different each year. Then, also the distribution along the year of the data can be different. For example, due to the rise in the summertime HCHO concentration, if frequent clouds caused few TROPOMI observations collected in the summer of a given year, the mean annual concentration of such a year could be smaller than the mean of the other year characterized by more summer HCHO observations. This confounding factor complicated the interpretation of HCHO changes."

l246: "sensitivity" please rephrase or explain.
R-> We explained it by adding the following discussion:
(l329-338)
"Traditionally, the ozone production regime is considered to be VOC-limited when this ratio is lower than 1, NOx-limited when it is higher than 2, while ozone is expected to be in the transition regime when the values are in the range 1–2 (Duncan et al., 2010; Ryan et al., 2020). Although several studies used this ratio to infer O3 sensitivity to NOx and VOCs by using observations from satellite and ground-based instruments (Duncan et al., 2010; Jin et al., 2017; Schroeder et al., 2017; Irie et al., 2021), some limitations still exist. Assuming the transition region lies within the range 1–2 (Duncan et al., 2010) could not be valid at global levels, and it could be necessary to compute it depending on the region (Schroeder et al., 2017). Moreover, the ratio has an altitude dependence (e.g., Jin et al., 2017; Schroeder et al., 2017). While seasonal variations and trends in the columnar HCHO/NO2 ratio (i.e., based on satellite observations) generally match the ratio computed with in situ observations, magnitudes are often different due to different vertical distributions of HCHO and NO2 (Ryan et al., 2020)."

l258: "recovered" A strange word for PBL ozone. "increased" would be better.
R-> Agreed!

l270: "assimilate satellite observations of tropospheric NO2" CAMS is adjusting concentrations, which implies that the impact of the assimilation is expected to be relatively short, and a short range (12h or 1 day) forecast is expected to differ only slightly from a run without NO2 satellite data assimilation.
R-> We agree. Here, we added the reference for these CAMS products i.e., Innesset al., 2019

What kind of CAMS product was used? Is it the analysis or the short range forecast? (may be good to mention this in 2.1.6)
R -> As mentioned in 2.1.6, we used the "CAMS global reanalysis (EAC4)" product (not the "CAMS global atmospheric composition forecasts" product).

l302: Figure 5c is a bit unclear. What are the steps between the red and black contours? Is it OMI (suggested by the caption) or TROPOMI (suggested by the text) based? It may be useful to introduce a separate figure for the 5-c panel.

R-> We apologize for the confusing caption. In the revised manuscript, we made clear the text of the caption, and we specified the steps of the isolines. Moreover, please note that the details of the method to build Fig. 5c have been more explicitly described in the Methods Section.

The figure is based on TROPOMI NO2 observations in 2019-2020. Black contours show the (absolute) NO2 composite differences of high wind speed days minus low wind speed days; see our reply above at l242. Red contours show the NO2 composite differences of Sunday minus weekdays. Red and black isolines show the same values but somewhat different patterns (contours for both red and black solid lines read as follows: -1,-2,-3,-4,-5,-6 $x10^{15}$ molec./cm$^2$. Then, the black dashed lines show also positive changes (0.5, 1 $x10^{15}$ molec./cm2)).

We want to preserve it as a panel within Fig. 5 to easily compare it with the OMI-based maps, so we have provided a high-resolution figure for the revised manuscript, allowing for reading the details.

l411-419: The absence of an ozone weekend effect is indeed somewhat surprising. I was wondering if a more clear signal is found when only winter or summer months are selected? One expects more titration in winter, and more formation in summer.

R -> Yes, we agree. However, unfortunately there are no MAX-DOAS ozone data in winter (see Sect. 2.1.1). Although, as detailed in the main text, previous surface observations showed the weekend effect in Tokyo, our MAX-DOAS O3 partial column observations did not show the potential weekend effect for various reasons as follows:

- The ozone profile is different from the other trace gases; in contrast to NO2, which strongly decreases its concentration with altitude, ozone concentration does not decrease with altitude.
- The ozone weekend effect at the surface level is usually 10 % (Sadanaga et al., 2012) and is much less evident than NO2. As suggested by ozonesonde observations (Fig. 5e), ozone changes due to titration maximize at the surface and tend to reduce shortly at h > 0.5 km. Since MAX-DOAS O3 partial column observations sampled the 0-1 km layer, the effect tends to disappear in our data.
- More titration is expected in winter, but MAX-DOAS O3 observations were unavailable this season (Sect. 2.1.1).
- Finally, the number of MAX-DOAS daily ozone samples is generally smaller than the other trace gases.

In the revised manuscript, we rephrased the text highlighting the points above.

Section 4 discussion: This section lists the main conclusions, but could be extended by listing shortcomings and with suggestions for future improvements and outlook on new datasets to be explored in the future (e.g. new satellite missions).

R -> Thank you the suggestion. We added the following discussion including shortcomings, future improvements and outlook on new satellite-based datasets:

(l589-601)

"Although not explicitly mentioned in the previous discussion, an implicit assumption of our study relies on the fact that satellite observations available only around midday are representative of daily changes computed, for example, by hourly observations. Although we provide evidence that this is likely the case, data from new geostationary satellites (e.g., Geostationary Environment Monitoring Spectrometer on board the Geostationary Korea Multi-Purpose Satellite 2) are expected to shed some further light on this issue.

A further shortcoming is the scarcity of reliable tropospheric ozone datasets to complement the satellite-based spatial distribution achieved with NO2 and HCHO observations. Despite the recent progress (Shen et al., 2019), OMI O3 only has some low sensitivity to the boundary layer, and this would make challenging any analysis over the investigated region (past studies found some correlation with the actual surface ozone in China, where tropospheric ozone is much larger, Shen et al., 2019). TROPOMI is expected to improve this capability soon, but its ozone dataset is currently limited to tropical latitudes.

Finally, it is worth mentioning the potential impact of the rebound of the long-range transport of pollutants after the Chinese economic recovery from the COVID-19 pandemic (Itahashi et al., 2022) on the current pollution within the Kanto region will deserve further investigation."

Additional references

- Barré, J., Petetin, H., Colette, A., Guevara, M., Peuch, V.-H., Rouil, L., Engelen, R., Inness, A., Flemming, J., Pérez García-Pando, C., Bowdalo, D., Meleux, F., Geels, C., Christensen, J. H., Gauss, M., Benedictow, A., Tsyro, S., Friese, E., Struzewska, J., Kaminski, J. W., Douros, J., Timmermans, R., Robertson, L., Adani, M., Jorba, O., Joly, M., and Kouznetsov, R.: Estimating lockdown-induced European NO2 changes using satellite and surface observations and air quality models, Atmos. Chem. Phys., 21, 7373–7394, https://doi.org/10.5194/acp-21-7373-2021, 2021.
- Brancher M., Increased ozone pollution alongside reduced nitrogen dioxide concentrations during Vienna's first COVID-19 lockdown: Significance for air quality management, Environmental Pollution, Volume 284, 2021, 117153, https://doi.org/10.1016/j.envpol.2021.117153.
- Duncan, B. N., Yoshida, Y., Olson, J. R., Sillman, S., Martin, R. V., Lamsal, L., Hu, Y. T., Pickering, K. E., Retscher, C., Allen, D. J., and Crawford, J. H.: Application of OMI observations to a space-based indicator of NOx and VOC controls on surface ozone formation, Atmos. Environ., 44, 2213–2223, https://doi.org/10.1016/j.atmosenv.2010.03.010, 2010.
- Fioletov, V., McLinden, C. A., Griffin, D., Krotkov, N., Liu, F., and Eskes, H.: Quantifying urban, industrial, and background changes in NO2 during the COVID-19 lockdown period based on TROPOMI satellite observations, Atmos. Chem. Phys., 22, 4201–4236, https://doi.org/10.5194/acp-22-4201-2022, 2022.
- Guevara M., O. Jorba, A. Soret, H. Petetin, D. Bowdalo, K. Serradell, C. Tena, H. Denier van der Gon, J. Kuenen, V.-H. Peuch, C. Pérez García-Pando, Time-resolved emission reductions for atmospheric chemistry modelling in Europe during the COVID-19 lockdowns, Atmos. Chem. Phys., 21 (2021), pp. 773-797, 10.5194/acp-21-773-2021
- Itahashi, S., Yamamura, Y., Wang, Z. et al. Returning long-range PM2.5 transport into the leeward of East Asia in 2021 after Chinese economic recovery from the COVID-19 pandemic. Sci Rep 12, 5539 (2022). https://doi.org/10.1038/s41598-022-09388-2
Jin, X., Fiore, A. M., Murray, L. T., Valin, L. C., Lamsal, L. N., Duncan, B., Boersma, K.F., De Smedt, I., Abad, G.G., Chance, K., and Tonnesen, G. : Evaluating a space-based indicator of surface ozone-NOx-VOC sensitivity over midlatitude source regions and application to decadal trends. J. Geophys. Res., 122(19), 10,439-410,461, https://doi.org/10.1002/2017JD026720, 2017
- Ju M. J., J. Oh, Y. H. Choi et al., Changes in air pollution levels after COVID-19 outbreak in Korea, Science of the Total Environment 750, 141521, https://doi.org/10.1016/j.scitotenv.2020.141521, 2021
- Laughner J. L. et al., Societal shifts due to COVID-19 reveal large-scale complexities and feedbacks between atmospheric chemistry and climate change, 2021, 118 (46) e2109481118, https://doi.org/10.1073/pnas.2109481118
- Le T, Wang Y, Liu L, Yang J, Yung YL, Li G, Seinfeld JH. Unexpected air pollution with marked emission reductions during the COVID-19 outbreak in China. Science. 2020, 369(6504):702-706. doi: 10.1126/science.abb7431
- Levelt, P. F., Stein Zweers, D. C., Aben, I., Bauwens, M., Borsdorff, T., De Smedt, I., Eskes, H. J., Lerot, C., Loyola, D. G., Romahn, F., Stavrakou, T., Theys, N., Van Roozendael, M., Veefkind, J. P., and Verhoelst, T.: Air quality impacts of COVID-19 lockdown measures detected from space using high spatial resolution observations of multiple trace gases from Sentinel-5P/TROPOMI, Atmos. Chem. Phys., https://doi.org/10.5194/acp-2021-534, 2022 (in press)
- Liu et al., Diverse response of surface ozone to COVID-19 lockdown in China, Science of the Total Environment, 789, 147739, 2021, https://doi.org/10.1016/j.scitotenv.2021.147739
- Liu, F., Beirle, S., Zhang, Q., Dörner, S., He, K., and Wagner, T.: NOx lifetimes and emissions of cities and power plants in polluted background estimated by satellite observations, Atmos. Chem. Phys., 16, 5283–5298, https://doi.org/10.5194/acp-16-5283-2016, 2016.
- Shakil, MH, Munim, ZH, Tasnia, M, Sarowar, S. 2020. COVID-19 and the environment: A critical review and research agenda. Science of the Total Environment 745. DOI: http://dx.doi.org/10.1016/j.scitotenv.2020.141022.
- Shen, L., Jacob, D. J., Liu, X., Huang, G., Li, K., Liao, H., and Wang, T.: An evaluation of the ability of the Ozone Monitoring Instrument (OMI) to observe boundary layer ozone pollution across China: application to

2005–2017 ozone trends, Atmos. Chem. Phys., 19, 6551–6560, https://doi.org/10.5194/acp-19-6551-2019, 2019.

- Sillman, S.: The use of NOy , H2 O2 , and HNO3 as indicators for Ozone-NOx -Hydrocarbon sensitivity in urban Locations, J. Geophys. Res. Atmos., 100, 14175–14188, https://doi.org/10.1029/94jd02953, 1995
- Surl, L., Palmer, P. I., and González Abad, G.: Which processes drive observed variations of HCHO columns over India?, Atmos. Chem. Phys., 18, 4549–4566, https://doi.org/10.5194/acp-18-4549-2018, 2018
- Takashima H, Irie H, Kanaya Y, Shimizu A, Aoki K, Akimoto H (2009) Atmospheric aerosol variations at Okinawa Island in Japan observed by MAX-DOAS using a new cloud screening method. J Geophys Res 114(D18):D18213. https://doi.org/10.1029/2009JD011939
- Verhoelst, T., et al.: Ground-based validation of the Copernicus Sentinel-5P TROPOMI NO2 measurements with the NDACC ZSL-DOAS, MAX-DOAS and Pandonia global networks, Atmos. Meas. Tech., 14, 481–510, https://doi.org/10.5194/amt-14-481-2021, 2021.

---

## Author Response (AR2)

**Reply to Reviews of "Peculiar COVID-19 effects in the Greater Tokyo Area revealed by spatiotemporal variabilities of tropospheric gases and light-absorbing aerosols" by Damiani et al.**

We included some additional information to the manuscript as requested by the Reviewer, but there are no relevant changes. Please note that below the Reviewer refers to line numbers of the revised tracked changes manuscript

We thank the Reviewer for reading our paper and providing constructive comments. We have incorporated the reviewer suggestions into the revised manuscript and provided point-by-point responses to each comment below.
In the following, we first report the referee comments (in black), then we provide our responses (in red).

The authors have addressed most of my comments and the manuscript is greatly improved. I believe the manuscript would benefit from further edits, and so I recommend this paper for publication after minor revisions.
R -> We thank the Reviewer for reviewing our manuscript once again. Below we addressed the additional comments.

Abstract – Total column is typically used to denote the total atmospheric column and includes the stratospheric component of species. I understand that the authors want to differentiate the total tropospheric column from the partial column and so need to carefully use these terms. One suggestion is to have all instances of 'total column' as 'total tropospheric column'. In the abstract and in the manuscript, the statement can be 'total and partial tropospheric column'.
R -> We agree. When possible, we now included the term "tropospheric".

L65 – Now that the authors mention about 'tropospheric ozone', it would be good to point out early in the manuscript why they do not use these in their analysis.
R -> We agree. Following your advice, we now stated early (in Section 2.1.3) that "Since OMI O3 only has some low sensitivity to the boundary layer while TROPOMI O3 is currently limited to tropical latitudes, we did not use satellite-based O3 datasets in this study."

L87-88 Should be 'Increases in surface O3'.
R -> OK! We changed the sentence as suggested.

L215-225 Would it be possible to comment on the minor differences between the NO2 product versions? Are the differences between the HCHO product versions also minor?
R -> The minor differences between TROPOMI NO2 versions 1.2.x. and 1.3.x affected only a small fraction of the observations. They were caused by improvements in the FRESCO-S algorithm devoted to retrieving cloud information. Indeed, since version 1.3.x., to avoid non-physical cloud fraction and pressure values, when the top of atmosphere reflectance is lower than expected, the surface albedo is reduced to match the top-of-atmosphere reflectance (Van Geffen et al., 2021). A further change in FRESCO is the treatment of very high cloud fractions (Van Geffen et al., 2021). Nevertheless, we excluded observations retrieved under these conditions. On the other hand, version 2.1.3 of the TROPOMI HCHO includes various improvements compared to the previous version 1.1.x., such as a new surface albedo retrieval algorithm, the adoption of new OCRA cloud-free maps, and the correction of some QF values over snow/ice regions (the latter did not affect the investigated area). More information can be found in the product read-me file and recent validation activities (De Smedt et al., 2021). We included this additional information in Section 2.1.2.

L235-240 Is it possible to calculate how much data you may lose and what would be the impact on the mean NO2 columns if you keep the same cloud threshold for both OMI and TROPOMI? I understand it maybe beyond the scope of the present study and so the authors can also reference to existing literature where the impact may have been assessed.
R -> Figure R1 shows the reduction in the number of OMI observations resulting from various cloud fraction (CF) thresholds (black) based on the long records of OMI NO2 data (2005-2020) recorded over Chiba station (e.g., for CF = 1, we used all data). Here, for sake of completeness, we also included the very uncertain observations recorded under high CF (i.e., CF > 0.5, shaded area). OMI observations reduce roughly linearly within the 0.3-0.9 CF range. Then, larger reductions occur for CF < 0.3. Taking a threshold of 0.3 (as in this study, dashed line) removes about 50% of the OMI observations while keeping the same threshold as that used for TROPOMI (i.e., 0.2) further reduces them by about 10%. Moreover, the overall tendency of the corresponding changes in the OMI NO2 column with the CF thresholds (red) shows that the NO2 column remains roughly constant for CF < 0.4. More information on the variation of OMI NO2 as a result of modifying screening criteria can be found in Compernolle et al. (2020). We now included some additional information in Section 2.1.3.

[Figure]

Fig. R1 - OMI NO2 data in 2005-2020 recorded over Chiba station showing changes in the number of OMI observations (black) and NO2 amount (red) as a function of various cloud fraction thresholds. Dashed line: OMI cloud fraction threshold used in this study (CF = 0.3).

Also, a brief comment on why OMI HCHO was not used in this work?
R -> In this study, we used OMI NO2 to evaluate the average weekly changes and, secondary, to confirm decreasing trends. Since biogenic emissions dominate HCHO, evident weekly cycles are not expected in OMI HCHO. Moreover, a recent study (De Smedt et al., 2021) has already shown the comparison between our full-time series of MAX-DOAS HCHO and OMI HCHO. So, we did not use OMI HCHO.

L580-585 This section can be strengthened. Is there a seasonality in the HCHO/NO2 weekly cycle? And a brief discussion is warranted if the observed differences are significant.
R -> Overall, due to the seasonal variations in NO2 and HCHO concentrations, the HCHO/NO2 ratio also shows significant seasonality, with a large ratio in summer compared to the other seasons (Irie et al., 2021). Nevertheless, the frequent cloudy conditions in the late-sprint to summer period and the limited temporal extension of the dataset prevent to evaluate seasonal differences in its weekly cycle. We included some additional information in Section 3.3.1.

L667-670 Why not perform a quick analysis to compare the NO2 from TROPOMI to that from GOME2 and strengthen this implicit assumption?
R -> Figure S5 (in the supplement), based on MAX-DOAS observations, already showed that changes observed around the satellite overpass (right column) were roughly representative of the daily changes (left column) at our location. As stated in the discussion, further evidence based on satellite observations would require geostationary observations. Comparing TROPOMI observations (with an early afternoon overpass) and GOME2 observations (with a mid-morning overpass) would hardly provide additional insights as the analysis, to be carried out over a region with a heterogeneous NO2 distribution, would be based on two different instruments characterized by very different spatial resolution (i.e., GOME 2: 40 km x 80 km vs. TROPOMI: 3.5 km x 5.6 km) and with a limited temporal resolution. We now explicitly mentioned Fig. S5 when mentioning this assumption.

Figure 1 Would it be better to also compute these changes for normalised mobility? We may be seeing a lower change in mobility for Japan because of a higher baseline, I presume.
R -> Please note that Google mobility data are already normalized to the baseline (Jan 3- Feb 6, 2020), and the baseline data is not provided. For the sake of clarity, we prefer keeping the original Figure 1.

New references
Compernolle, S. et al.: Validation of Aura-OMI QA4ECV NO2 climate data records with ground-based DOAS networks: the role of measurement and comparison uncertainties, Atmos. Chem. Phys., 20, 8017–8045, https://doi.org/10.5194/acp-20-801 7-2020, 2020.